# Nonlinear wave evolution with data-driven breaking

D. Eeltink ⬛ [1,2✉], H. Branger[3], C. Luneau[3], Y. He ⬛ [4], A. Chabchoub ⬛ [4,5,6], J. Kasparian ⬛ [7],
T. S. van den Bremer[2,8] & T. P. Sapsis[1✉]

Wave breaking is the main mechanism that dissipates energy input into ocean waves by wind and transferred across the spectrum by nonlinearity. It determines the properties of a sea state and plays a crucial role in ocean-atmosphere interaction, ocean pollution, and rogue waves. Owing to its turbulent nature, wave breaking remains too computationally demanding to solve using direct numerical simulations except in simple, short-duration circumstances. To overcome this challenge, we present a blended machine learning framework in which a physics-based nonlinear evolution model for deep-water, non-breaking waves and a recurrent neural network are combined to predict the evolution of breaking waves. We use wave tank measurements rather than simulations to provide training data and use a long short-term memory neural network to apply a finite-domain correction to the evolution model. Our blended machine learning framework gives excellent predictions of breaking and its effects on wave evolution, including for external data.

[1] Department of Mechanical Engineering, Massachusetts Institute of Technology, Cambridge, MA, United States. [2] Department of Engineering Science, University of Oxford, Oxford, UK. [3] Aix-Marseille University, CNRS, Centrale Marseille, IRPHE, Marseille, France. [4] Centre for Wind, Waves and Water, School of Civil Engineering, The University of Sydney, Sydney, NSW, Australia. [5] Disaster Prevention Research Institute, Kyoto University, Kyoto, Japan. [6] Hakubi Center for Advanced Research, Kyoto University, Kyoto, Japan. [7] Group of Applied Physics and Institute for Environmental Sciences, University of Geneva, Geneva, Switzerland. [8] Faculty of Civil Engineering and Geosciences, Delft University of Technology, Delft, The Netherlands. ✉email: eeltink@mit.edu; sapsis@mit.edu

The importance of wave breaking, the main mechanism that dissipates energy input into ocean waves, is twofold. First, wave breaking provides an upper limit to how tall or steep waves can become, thereby limiting the steepening effects of winds[1], currents[2], crossing seas[3], refraction by bathymetry[4], abrupt depth transitions[5,6], and nonlinear focusing[7–9]. Second, wave breaking itself plays a crucial role in important physical processes, such as the transport and dispersion of surface pollution including plastic debris and oil[10], the energy, momentum, and mass fluxes in ocean-atmosphere interactions[11,12] with climate applications such as atmosphere-ocean $CO_2$ exchange[13,14], and the formation of rogue waves[15].

Despite its ubiquity and importance, no satisfying models for wave breaking exist. In potential-flow models, such as (variants of) the nonlinear Schrödinger (NLS) equation[7,16] and higher-order spectral methods (HOSM)[17], the effects of breaking and the vorticity the breaking induces are ignored as a direct consequence of the potential-flow assumption; they must be re-introduced through reduced-form breaking terms added to the model in ad-hoc fashion[18–21]. Such reduced-form breaking terms have been validated for highly simplified cases, but not for realistic broad-banded spectra, and require parameter tuning, yielding them unfit for prediction. On a larger scale, spectral wave models, such as WaveWatch III, include wave breaking through empirically determined dissipation modules[22–24], but do not resolve individual waves.

Direct numerical simulations (DNS) of the Navier–Stokes equations have to be performed in 3D to explicitly resolve turbulence and have only recently been used to capture the entire pre- and post-breaking of only a single wave in 3D[25], although at very high computational cost. The 2D approximation[26,27] offers a promising prospect but remains so computationally demanding that only a very narrow region in space and time can be studied. In short, none of the three methods to describe breaking discussed above can be used in practical wave-resolving forecasting.

In other related fields, machine learning (ML) has recently been used to complement physical models with great success. In climate modeling, where the main uncertainty comes from estimating sub-grid processes, ML techniques can be used to emulate physical processes not resolved by the climate model[28]. In fluid dynamics, ML in conjunction with the Navier–Stokes equations can be used to obtain the pressure on the wall of an aneurysm from just an image of the velocity field[29]. Using ML, extreme event statistics in nonlinear dynamical systems can be correctly predicted from only a small number of samples[30]. In so-called blended machine learning specifically, a simplified physics-based model is combined with an ML algorithm to capture the physical processes missing from the simplified model. The algorithm is then trained on data to learn the full solution or ground truth. This approach has been successfully applied to spatiotemporally non-local turbulence closures[31] and particle trajectories in vortical flows[32]. The key ingredient is to include the memory of previous time steps in a recurrent neural network (RNN), allowing non-local representations by which the missing physical processes or "closure terms" can be parameterized. While ML studies have been performed in the context of wave breaking, these have focused on the detection and classification of breaking waves[33–35], rather than their prediction and simulation.

In this paper, we develop a blended machine learning framework to model wave breaking and its effects on the nonlinear evolution of ocean waves. We show that the framework can be used for wave-resolving forecasting of breaking waves. In doing so, we overcome the challenge of having to explicitly model the turbulent nature of wave breaking. In the framework we develop, we use the viscous modified NLS (MNLS) equation[7,16,36,37] as the

physics-based model for non-breaking waves and wave tank measurements to serve as the ground truth.

Several mechanisms influence the evolution of waves, both in a tank and in the ocean. These include wind forcing, dispersion, nonlinear interactions, and dissipation as a result of kinematic viscosity, tank side-walls, and wave breaking. The MNLS is a canonical wave model that can predict the nonlinear and dispersive evolution of the (complex) envelope $a$ of the free surface of unidirectional water waves accurately[38] and at a very low computational cost, provided the waves are in deep water, do not break, there is no energy input from wind, and dissipative effects other than the kinematic viscosity are excluded. In non-dimensional form, the MNLS we use reads:

$$\underbrace{\frac{\partial a}{\partial \xi} + i\frac{1}{2}\frac{\partial^2 a}{\partial \tau^2} + ia|a|^2}_{\text{NLS}} = \epsilon \underbrace{\left(8|a|^2\frac{\partial a}{\partial \tau} + 2a^2\frac{\partial a^*}{\partial \tau} + 2ia\mathcal{H}\left[\frac{\partial |a|^2}{\partial \tau}\right]\right)}_{\text{higher-order terms}}$$
$$\underbrace{-\delta_0 a - i\delta_1\frac{\partial a}{\partial \tau} - \delta_2\frac{\partial^2 a}{\partial \tau^2}}_{\text{viscous damping}}, \tag{1}$$

where $\xi$ is dimensionless space, $\tau$ dimensionless time, the steepness $\epsilon = A_0 k_0 \sqrt{2}$ with $A_0$ the characteristic surface elevation amplitude and $k_0$ the carrier wave number, the * symbol denotes the complex conjugate, $\mathcal{H}$ a Hilbert transform, and the zeroth, first and second-order viscous contributions have coefficients $\delta_0 = 1/(2\epsilon)\nu$, $\delta_1 = 5\nu/2$, $\delta_2 = 5\epsilon\nu$ respectively, where $\nu = T_0 4k_0^2 \tilde{\nu}$ with $T_0$ the carrier wave period and $\tilde{\nu} = 1.00 \times 10^{-6}$ m²/s the kinematic viscosity.

From the listed mechanisms, the NLS terms[7] in eq. (1) capture the key physical processes (i.e., nonlinearity and dispersion) that are responsible for the Benjamin–Feir[39] or modulational instability (MI), which characterizes the evolution of non-breaking deep-water surface gravity waves. The higher-order terms[16] account for asymmetries in the spectrum and allow for steep (narrow-banded) waves to be modeled accurately[38]. The MNLS is derived from the water-wave equations using a perturbation expansion in steepness and a slowly varying envelope approximation[16,40]; as such, its validity is compromised for very broad-banded spectra. We emphasize that the spatial-evolution version of the MNLS we use (i.e., eq. (1)) captures linear dispersion exactly even for broad-banded spectra, whereas temporal-evolution versions require additional higher-order (spatial) derivatives[41–44]. The bandwidth limitations of eq. (1) arise because of the combination of broad bandwidth and nonlinearity. The viscous damping terms in eq. (1) account for the kinematic viscosity, and more importantly, restrict the growth of spurious high-frequency waves when the MNLS reaches unphysical amplitudes[36,37]. We do not account for side-wall friction, which does not have a significant effect in our experiments. Furthermore, we study the effect of wave breaking in isolation and therefore omit wind forcing[45,46].

For the blended framework, high-fidelity turbulence-resolving direct numerical simulations remain too computationally expensive as a method to generate training data. We instead use measurements of breaking waves in a wave tank as the ground truth (see Methods). We obtain a training data set consisting of three wave types. Modulated plane waves (Wave Category I) are the simplest idealized example of breaking waves, reaching their maximum amplitude due to MI. Dispersively focused irregular waves (Wave Category II) provide a more realistic representation of the ocean[47], and can reach a breaking amplitude due to focusing on the phases of the different-frequency wave components. Modulated plane waves and focused irregular waves (Wave Categories I-II) are chosen deliberately as they are the only two

wave types for which breaking can be clearly and non-controversially detected in the spectrum. Finally, irregular waves with random phases (Wave Category III) are closest to a (unidirectional) sea state, where the number of breaking waves is sporadic, depending on the significant wave height, and breaking is harder to detect. The data set is split into training, validation, and test sets, of which only the first two are used in the training and training optimization process of the model. In addition, the model is only trained on segments of the total propagation length of the experiments.

Although measurements have the added advantage of being closer to the ground truth than any model (measurement errors notwithstanding) thus capturing the relevant physical processes more completely, data is only available at a finite number of measurement locations, where wave gauges are positioned, and not at each solver step. While the wave envelope varies sufficiently slowly and can be interpolated, the phase cannot (see SI section 1). To overcome the requirement of conventional machine learning methods[31,32,48] to have the full ground truth at each solver step, we develop a finite-domain machine learning (FDML) correction, which applies an RNN over multiple solver steps. The RNN allows the network's memory states to detect signals, such as steepening or spectral broadening, as predictors of wave breaking. In addition, it allows future information of the simplified model (MNLS) to influence the correction at earlier time steps. The algorithm is thus non-local and non-causal, partly explaining its efficiency. The FDML framework we develop can be used in other applications in which only part of the ground truth is available, such as in optical fibers, in which it is notoriously difficult to measure phase[49].

## Results

**Wave Category I: modulated plane waves.** Figure 1 shows results for the canonical case of a modulated (or perturbed) plane wave (Wave Category I), consisting of a carrier wave perturbed by upper and lower sidebands, resulting in dynamics that are mainly determined by only three spectral modes. Due to modulational instability[7,39], the spectral sidebands grow, and the spectrum broadens. In the time domain, this corresponds to an amplification of the amplitude. In the MNLS simulation (Fig. 1a, e), an approximate recurrence back to the initial condition occurs, known as the Fermi–Pasta–Ulam–Tsingou (FPUT) recurrence[50–52]. In the experiment (Fig. 1c), the amplitude amplification leads to breaking at $\xi = 3.42$, most notably resulting in the lower sideband becoming dominant (Fig. 1g)[8,53,54].

We note that existing breaking models, such as the steepness threshold model by[19] and the nonlinear dissipation term in[18] correctly predict a spectral downshift for a modulated plane wave, although they have not been compared to experimental data therein (see Supplementary Information for a comparison with our results). The kinetic energy equation-based model[20,21] also gives good qualitative agreement for the downshift with modulated plane wave experiments but requires parameter tuning.

Our MNLS + FDML model correctly predicts the permanent downshift of the peak (Fig. 1f) and consequently of the spectral mean. The large amplitudes predicted by the MNLS alone (Fig. 1a) leading to breaking and amplitude reduction in experiments (Fig. 1c) are capped correctly (Fig. 1b). The dispersive spread of the measurements (Fig. 1c) at long distances is captured well by the MNLS+FDML model, although the light-colored 'valleys' of the envelope in the experiments are somewhat deeper and broader.

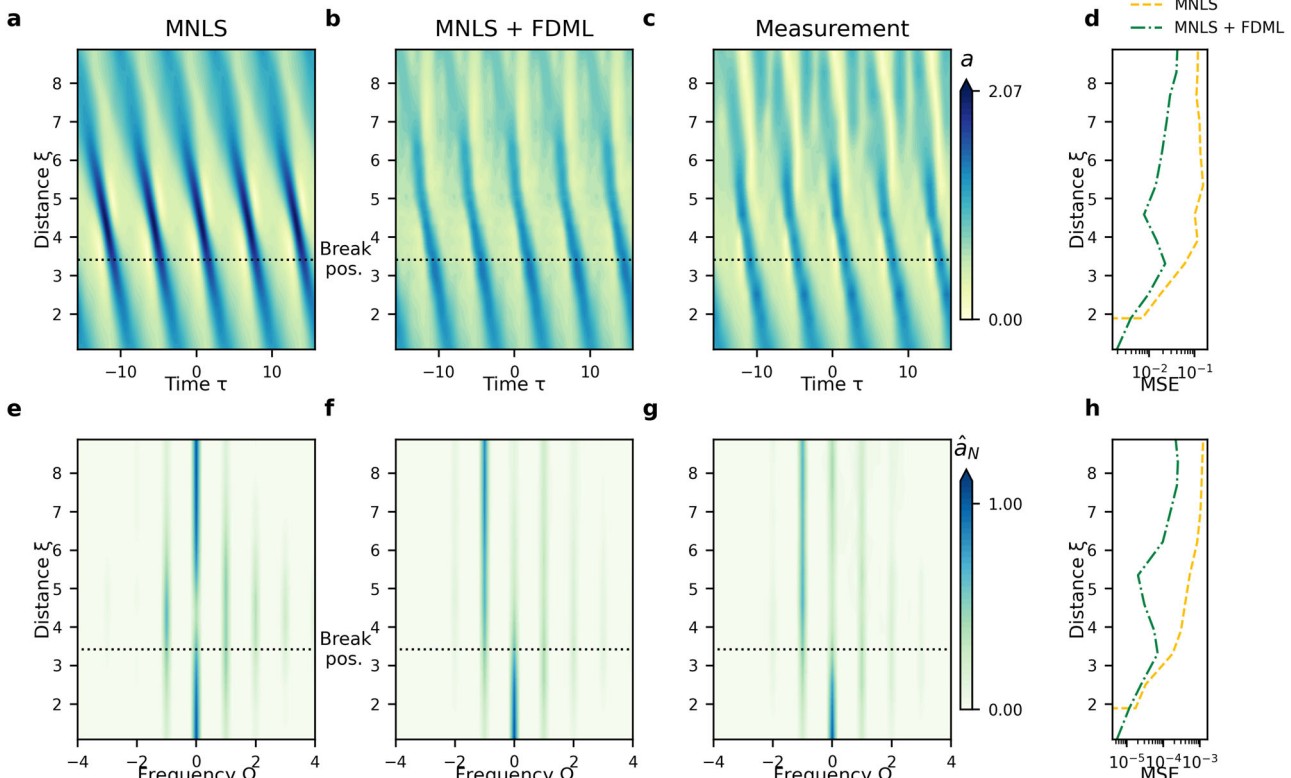

**Fig. 1 Example result (not used for training) for the spatial evolution of a modulated plane wave (Wave Category I) showing wave breaking at $\xi = 3.42$, as indicated by the black dotted line. a–d** Time domain. Color bar indicates surface envelope amplitude $a$ (see eq. (1)). **a** MNLS simulations. **b** MNLS + FDML simulations. **c** Measurements. **d** Mean squared error (MSE) at each wave gauge between measurements and MNLS or MNLS + FDML simulations. **e–h** Frequency domain, similar panel configuration as in the time domain, with the color bar indicating the magnitude of the amplitude spectrum normalized by the maximum of the initial condition ($\hat{a}_N$).

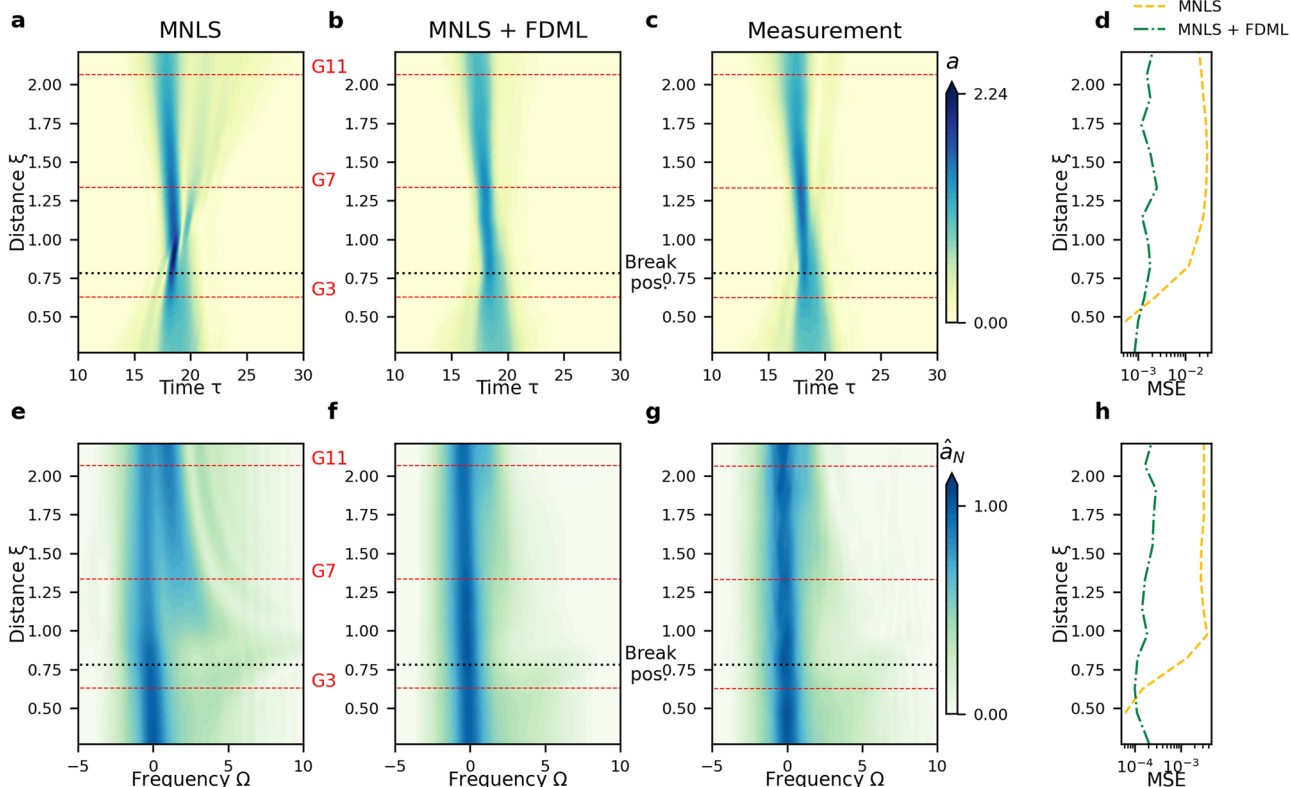

**Fig. 2 Example result (not used for training) for the evolution of focused irregular waves (Wave Category II) showing wave breaking at $\xi = 0.78$, as indicated by the black dotted line. a–d** Time domain. Color bar indicates surface elevation envelope $a$. **a** MNLS simulations. **b** MNLS + FDML simulations. **c** Measurements. **d** Mean squared error (MSE) at each wave gauge between measured envelope and MNLS and MNLS + FDML. **e–h** Frequency domain, similar panel configuration as in the time domain, with the color bar indicating the magnitude of the spectrum normalized by the maximum of the initial condition ($\hat{a}_N$). The horizontal red dashed lines correspond to the gauge locations examined in Fig. 4.

Importantly, our MNLS+FDML model does not apply an over-correction for non-breaking waves (see SI section 2).

**Wave Category II: dispersively focused irregular waves**. To examine wave fields that form a more realistic representation of the real ocean environment, we apply our MNLS + FDML framework to focused irregular waves created using a JONSWAP spectrum[47] as the input condition. The phases of the components of this spectrum are chosen so that they all come into phase (according to linear theory) at position $\xi_f$ in the tank, creating a steep wave that will break at $\xi_f$ or before.

Figure 2 shows the evolution of an example wave, breaking at $\xi = 0.78$. The measured envelope evolution differs from the MNLS prediction, and equally from a non-breaking wave, in several ways. First, in the time domain, the MNLS model (Fig. 2a) predicts a much higher amplitude around $\xi = 0.78$ than the measurement (Fig. 2c), as also manifested by the much higher steepness (Fig. 3c). Second, breaking causes a sudden loss of about 30% of the total energy (calculated as the norm of the envelope, i.e., $N(\xi) = \int \hat{a}^2 d\Omega$), as shown in Fig. 3a, unlike the MNLS prediction. This energy loss (not a redistribution across the spectrum) is due to the turbulent nature of breaking. Third, the MNLS predicts strong dispersive spreading for $\xi > 0.78$ (Fig. 2a, e) not present in the measurements (Fig. 2c, g). The additional high-frequency components present in the MNLS predictions at long distances (Fig. 2e) travel at slower speeds (Fig. 2a), in accordance with the linear surface gravity wave dispersion relation. The discrepancy in high-frequency components between MNLS and measurements is most clearly displayed in the time series at $\xi = 1.33$ (gauge 7, Fig. 4b) and at $\xi = 2.06$ (gauge 11,

Fig. 4c). Focusing in the time domain corresponds to a broadening of the spectrum at $\xi = 0.78$ in the MNLS simulations (Fig. 4d). In reality, these additional high-frequency components are quenched by breaking (comparing Fig. 2e to g and Fig. 4d to e, f). Finally, wave breaking induces a downshift of the spectral peak and the spectral mean[8,53–55]. The MNLS prediction does not account for the former (Fig. 4e, f), and even shows a slight upshift of the mean frequency, as shown in Fig. 3b, whereas the downshift in the measurement is dramatic.

For the focused irregular waves, the architecture and training procedure of the FDML algorithm is identical to that used for the modulated plane wave, but the training data set consists of focused wave groups with initial JONSWAP spectra. The MNLS +FDML model is able to reproduce accurately the measured evolution in both the time and the frequency domain. The significant improvement compared to the MNLS model is indicated by the mean squared error (MSE) evolution in Fig. 2d, h. Examining the effect of breaking, Fig. 2b shows the maximum amplitude of the envelope is attenuated to a value close to the measurement value, as corroborated by the sharp decline of the norm in Fig. 3a. In addition, the suppression of the dispersive spreading at long distances is correctly predicted. In the spectrum, Fig. 2f shows the spectral width at the breaking point is reduced. Figure 4e, f shows that the (slight) downshifting of the peak is correctly reproduced. The downshift of the mean frequency closely follows the measurements (Fig. 3b). As for modulated plane waves, for non-breaking focused irregular waves our MNLS + FDML model does not apply an over-correction. We refer to the Supplementary Information for an example of a non-breaking wave.

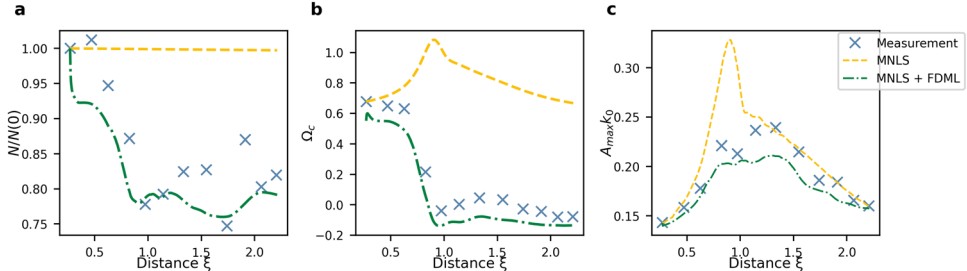

**Fig. 3 Evolution of summary parameters of the focused irregular waves (Wave Category II) shown in Figs. 2 and 4. a** Evolution of the norm $N/N_0$
**b** Evolution of the mean frequency $\Omega_c$. Note that $\Omega = 0$ corresponds to the central peak of the initial spectrum, which is lower than $\Omega_c$ for the initial
condition. **c** Evolution of the characteristic steepness $\eta_{max}k_0$.

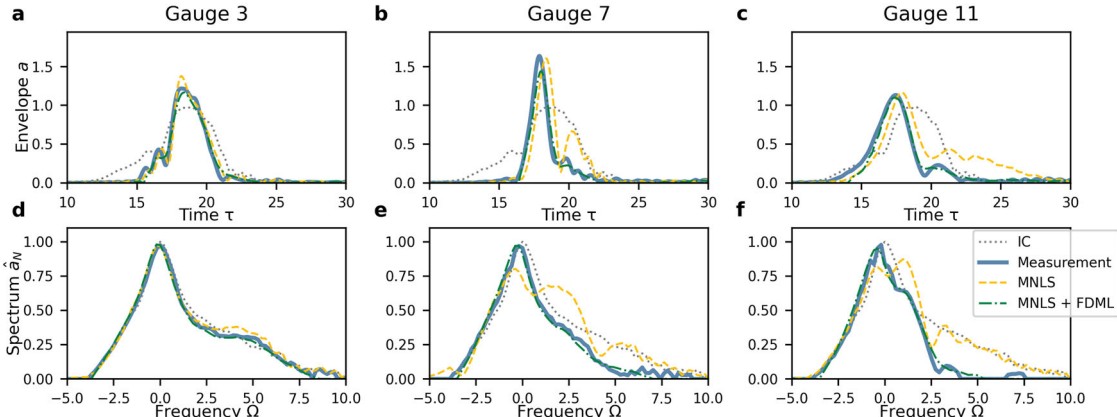

**Fig. 4 Envelopes and amplitude spectra of the focused irregular waves (Wave Category II) at different wave gauge locations indicated by the red-
dotted lines in Fig. 2. a–c** Time-domain envelopes. **d–f** Amplitude spectra. **a, d** Gauge 3. **b, e** Gauge 7. **b, e** Gauge 11. Initial condition (IC, dotted gray),
measurements (solid blue), MNLS simulations (dashed yellow), and MNLS + FDML simulations (dashed-dotted green).

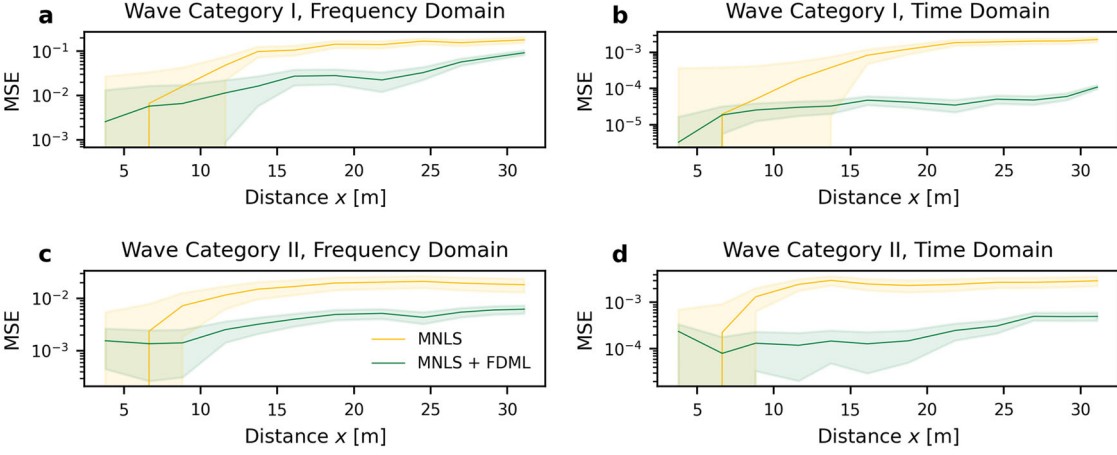

**Fig. 5 Mean squared error (MSE) relative to the ground truth for the MNLS + FDML model compared to the MNLS model, averaged over all
experiments from the test set (the set of data not used for training) that are breaking.** The shaded region corresponds to ±1 standard error across cases.
Data were shown for the full propagation length. Wave Category I in the time domain (**a**) and frequency domain (**b**) and Wave Category II in the time
domain (**c**) and frequency domain (**d**). We plot the dimensional distance $x$ on the horizontal axis, as the dimensionless distance $\xi$ is different for each
sample.

To assess the overall performance of the MNLS + FDML model,
Fig. 5 shows the mean squared error (MSE) for the full length of
propagation, averaged over the experiments in the test set, i.e., those
that were not used for training (see Supplementary Information for
all cases from the test set). The FDML correction is responsible for an
order-of-magnitude improvement compared to the MNLS predic-
tion. To verify the generality of our MNLS + FDML model, we also
predict wave breaking in an external data set of focused breaking

waves recorded in the Multifunctional Ship Model Towing Tank at
Shanghai Jiao Tong University, which has a much larger tank and
wave dimensions than our training set and find comparable results to
those reported here. Results are displayed in SI section 3.4.

**Wave Category III: random irregular waves.** Moving yet further
to a realistic random sea state, we investigate the ability of our

MNLS + FDML model to provide forecasts of (unidirectional) irregular waves based on a broad-banded JONSWAP spectrum with peak enhancement factor $\gamma = 3.3$, with random instead of focused phases (Wave Category II). In this category, both dispersion and MI can lead to wave steepening. Random sea state forecasts face two challenges. First, based on a finite-duration measurement at a point, random irregular waves can, as a result of dispersion, only be forecast over a finite so-called predictable region, which narrows as the time horizon of prediction increases or spectra become more broad banded[56]. Second, the signature of breaking is much less pronounced in the spectrum of (broad-banded) irregular waves. We therefore only compare the physical-space evolution. The training data generation for this method is detailed in SI section 5.

Because wave breaking in an irregular sea is generally sporadic, the dynamics (and the MSE) are dominated in large part by the waves that do not break. To assess the ability of the MNLS + FDML model to capture breaking, we select and examine wave groups from the test data set that are sufficiently steep, according to the criterion $\bar{a}k_0 > 0.28$, ensuring a high likelihood of breaking. These events are displayed in SI section 5.3.

Figure 6 shows the evolution of a representative example, with a breaking event occurring around $\xi = 1.1$ (gauge 7). The MNLS overestimates the amplitude, even predicting a non-physically large-amplitude indicated by the red circle in Fig. 6a, whereas the MNLS + FDML model corrects this over-estimation. For non-breaking waves, the performance of the MNLS+FDML model is comparable to that of the MNLS only, see SI Fig. 5.7. When there is a breaking event, the MNLS+FDML model outperforms the MNLS model, see Fig. 6d. The model has been trained on segments with a maximum length of half the propagation length in Fig. 6. Although the MNLS+FDML model has been trained on a broad-banded JONSWAP spectrum (peak enhancement factor $\gamma = 3.3$), it works equally well for a more narrow-banded JONSWAP spectrum ($\gamma = 6$, see SI Fig. 5.3).

## Discussion

We have demonstrated that the MNLS + FDML framework developed herein can predict the unidirectional evolution of the wave envelope and its spectrum for an arbitrary propagation length, even if wave breaking occurs along the way. This includes the correct prediction of dissipation of total energy by breaking, as well as the resulting suppression of higher frequencies and the downshift of the mean and peak frequency of the spectrum. The attenuation of the maximum amplitude of the surface elevation and the reduction in dispersive spreading are also correctly predicted. We have shown that our developed method is successful for modulated plane waves (Wave Category I), JONSWAP spectrum-based dispersively focused wave groups (Wave Category II), and broad-banded JONSWAP random irregular wave evolution (Wave Category III), increasing the degree of realism step by step. As we work with non-dimensional values for the wave amplitude and the carrier wave frequency, we expect that our results generalize well to other wave heights and wavelengths.

Our framework employs experimental data as the ground truth instead of a high-fidelity numerical model. Using measurements allows previously inaccessible physics to be included in the model, as opposed to just achieving a speed-up of the simulation of known physics by a more complex model[31,32,57–59]. This direct access does come at a price when considered in the context of the growing body of work[60–63] in which machine learning algorithms discover partial differential equations (PDEs), parts thereof, or their solutions. First, the ground truth is not always known in full at the model solver step, as required in an infinitesimal-domain blended model[31,32], because of the finite resolution of measurements and the difficulty interpolating all aspects of the ground truth information to the required time step (such as the phase in this paper) or because it is notoriously difficult to measure them (such as the phase in optics[49]). Second, a convergence of the ML algorithm to a global minimum is not guaranteed if the error between measurements and the full solution (a PDE or PDE term

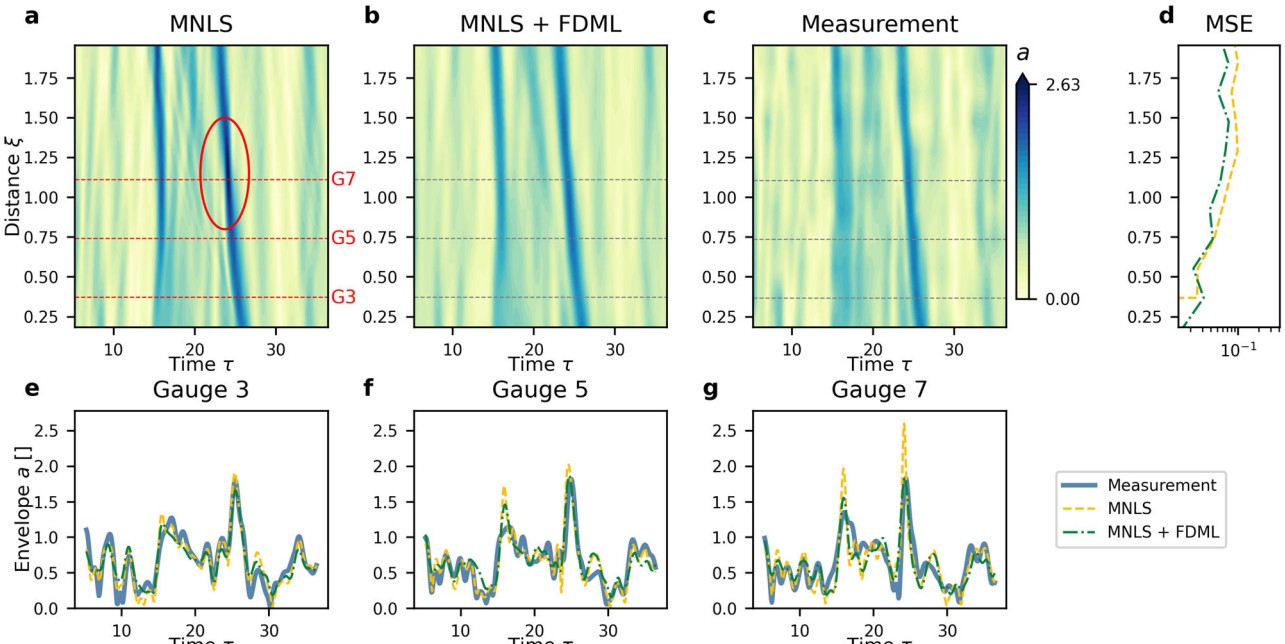

**Fig. 6 Example result (not used for training) for the evolution of the envelope of random irregular waves (Wave Category III), showing wave breaking at $\xi = 1.1$, as indicated by the red circle.** The color bar indicates surface elevation envelope $a$. **a** MNLS simulations. **b** MNLS + FDML simulations. **c** Measurements. **d** Mean squared error (MSE) at each wave gauge between measured envelope and MNLS and MNLS + FDML simulations. **e–g** Envelopes $a$ at the wave gauge locations indicated by the red-dotted lines in (**a**): measurements (solid blue), MNLS simulations (dashed yellow), and MNLS + FDML simulations (dashed-dotted green).

to be discovered) is due to missing physics instead of simply Gaussian noise added to synthetic data[60,61]. Third, when measurement data is only available in limited quantities, purely data-driven approaches are not successful, and a physics-based model is essential to compensate for the lack of data[61,64]. For instance, for optical fibers, the evolutionary dynamics can be described by a Neural Network instead of the NLS equation[65], when trained on large volumes of data. However, in the water-wave setting, obtaining such large quantities of wave tank experiments is prohibitively expensive.

While the finite-domain ML framework we have developed addresses some of these challenges, it has limitations. The evolution of the envelope over a finite domain strongly depends on the wave input, as the nonlinear interactions quickly mix the effects of all terms in the PDE. Therefore, over a finite domain, the difference in evolution with and without breaking has entangled in it the effects of both the wave type and its inherent nonlinear behavior and the breaking behavior. Consequently, the parameters that minimize the loss function of the network have different values for different wave types, and the FDML model cannot yet extrapolate from one wave type to another. If phase information at the solver step becomes available through either measurements or simulations, we envisage that the infinitesimal-domain method could remedy this limitation in future work as the effect of breaking on the spectrum and the nonlinear (non-breaking) spectral evolution itself then can become decoupled.

The main novelty of our work is the demonstration that the turbulent breaking process can be captured by a neural network with memory, and decoupled from the potential flow. As wave breaking is a complex phenomenon, the first step must be to select circumstances in which the signature of breaking is clearly detectable in both the time and frequency domain. Fulfilling this requirement, we choose Wave Categories I (modulated plane waves) and II (dispersively focused irregular waves), which are conventionally studied in the context of wave breaking[8,26,66–68]. Our results for Wave Category III (random irregular waves) act as a proof of concept for the applicability of our FDML model in wave-resolving forecasts of realistic random sea states. In this latter category, both dispersion and MI can steepen the waves.

To tackle more realistic sea states, we anticipate two future directions of development, distinguishing those that make use of the infinite-domain and those that make use of the finite-domain approach. In the finite-domain approach, the same approach as in this paper can readily be used for other combinations of models and measurement techniques, depending on the purpose of the study, such as deterministic wave forecasting, statistics, or extreme event detection. For each purpose, the blended approach as in this work can be applied: a model that captures the basic physics without breaking, a machine learning layer with memory, and measurements that track the evolution at finite intervals.

Before turning to the challenging integration with other effects such as wind forcing, the effect of directional spreading should be examined. While the unidirectional wave is a valid approximation for sea states dominated by swell, most realistic sea states have a degree of directional spreading. To generalize our approach to directional sea states, the 1D MNLS equation will need to be replaced by the 2D MNLS[41,43,69] or a model that directly describes the surface elevation (instead of the envelope) and is less restrictive on bandwidth, such as higher-order spectral methods (HOSM)[17,70], or the Zakharov equation[7,71]. For the surface elevation, crest slows down is an additional signature of incipient breaking, which can be used to test the validity of results[72]. Challenges will arise from dealing with the finite predictable region in 2D, from the boundaries of a finite spatial domain or a finite-size wave tank, and from the need for large quantities of spatially resolved data to be suitable for machine learning. For the

latter, different measurement techniques such as stereo-imaging[73,74] could be used. Finally, although the effect of small degrees of directional spreading on breaking is understood[75], breaking in crossing seas is not and maybe much less dissipative and amplitude-limiting[3].

For the infinitesimal-domain method, the first challenge in future work will be to identify and develop suitable, high-fidelity models that capture at least the most salient features of wave breaking, are able to do so for a range of wave types and conditions, and can generate a sufficient amount of training data without the excessive computational cost. Numerical models based on the Reynolds-averaged Navier–Stokes equation with a turbulence model are a promising candidate for this[76]. In addition, for this method, it will be crucial to guarantee stability such that the error of the predicted evolution as indicated in Fig. 7b, does not increase cumulatively with every step. To prevent such instability and non-physical results, hard physical constraints can be enforced on the output of the network, often implemented as some form of outer loop optimization[5,77–80] or added as a penalty in the cost function (so-called soft constraints).

In conclusion, we have developed a blended framework to predict the evolution of waves that includes the abrupt and turbulent process of wave breaking based on a canonical potential flow-based wave evolution equation. We foresee that being able to incorporate breaking effects in wave evolution equations in a simple way will help lift restrictions breaking has placed on the validity of models explaining the behavior of waves subject to winds[1], currents[2], crossing seas[3], refraction by bathymetry[4], abrupt depth transitions[5,6], and nonlinear focusing[7–9]. The experiment-based blended framework for wave breaking we have developed in this paper builds a suitable foundation for application to wave-resolving forecasting in real-world sea states, which should be the direction of future research.

## Methods

**Wave tank experiments.** For Wave Categories I (modulated plane waves) and II (focused irregular waves), ground-truth training data was generated in the 40 m long, 2.7 m wide wave facility at Aix Marseille University, France, by means of a piston wavemaker. Waves were measured by 12 wave gauges placed along the center-line of the tank with a sampling rate of 400 Hz. Experiments for Wave Category III (random irregular waves) were performed in the 30 m long, 1 m wide facility at the University of Sydney, Australia, again by means of a piston wavemaker. An array of eight wave gauges was moved along the tank for repeated measurements, to obtain a final number of 24 wave gauges positions spaced 0.83 m apart (covering 19.1 m). The gauges had a sampling rate of 32 Hz. See Fig. 7a for the experimental set-up.

**MNLS simulations and non-dimensionalization.** MNLS simulations were carried out by integrating eq. (1) numerically using a split-step Fourier method. The envelope constructed from the measured surface elevation $\eta(t)$ at any wave gauge can be used as an initial condition. The envelope can be constructed from the surface elevation by means of the Hilbert transform:

$$\tilde{a}(x,t) = (\eta(x,t) + i\tilde{\eta}(x,t))e^{-i(k_0 x - \omega_0 t)}, \tag{2}$$

where $\tilde{\eta}$ is the Hilbert transform, $\tilde{\eta} = \mathcal{H}[\eta] = \mathcal{F}^{-1}\left[-i\,\text{sign}(\omega)\mathcal{F}[\eta]\right]$, with $\mathcal{F}$ a Fourier transform and $\omega$ the angular frequency. To obtain a smooth envelope, bound modes are filtered from the measurement signal, and smoothing filters are applied. Quantities have been made dimensionless as $a = \tilde{a}/\tilde{a}_0$, $\tau = t'/T_0$, and $\xi = x/L_0$, where $t' = t - x/c_g$ is the group time scale with $c_g = \frac{1}{2}\left(g/k_0\right)^{-1/2}$ the linear group velocity in the deep-water limit, $T_0 = 1/(\omega_0\epsilon)$, $L_0 = 1/(2\epsilon^2 k_0)$, $\tilde{a}$ the dimensional envelope, $\tilde{a}_0$ its initial value, and $\epsilon = \tilde{a}k_0/\sqrt{2}$ the steepness.

**Wave Category I.** The modulated plane wave consists of a carrier wave seeded by upper and lower sidebands with modulation frequency $\Omega_M$, defined in dimensionless form as:

$$a(0,\tau) = \sqrt{b_0} + \sqrt{b_{-1}}e^{i(\Omega_M\tau + \psi)} + \sqrt{b_{+1}}e^{i(-\Omega_M\tau + \psi)}. \tag{3}$$

where $\sqrt{b_0}$, $\sqrt{b_{+1}}$, and $\sqrt{b_{-1}}$ are the amplitudes of the main mode and upper and lower sidebands, respectively, and $\psi$ the relative phase between the sidebands and the carrier mode. The amplitudes of the three modes of the initial condition are

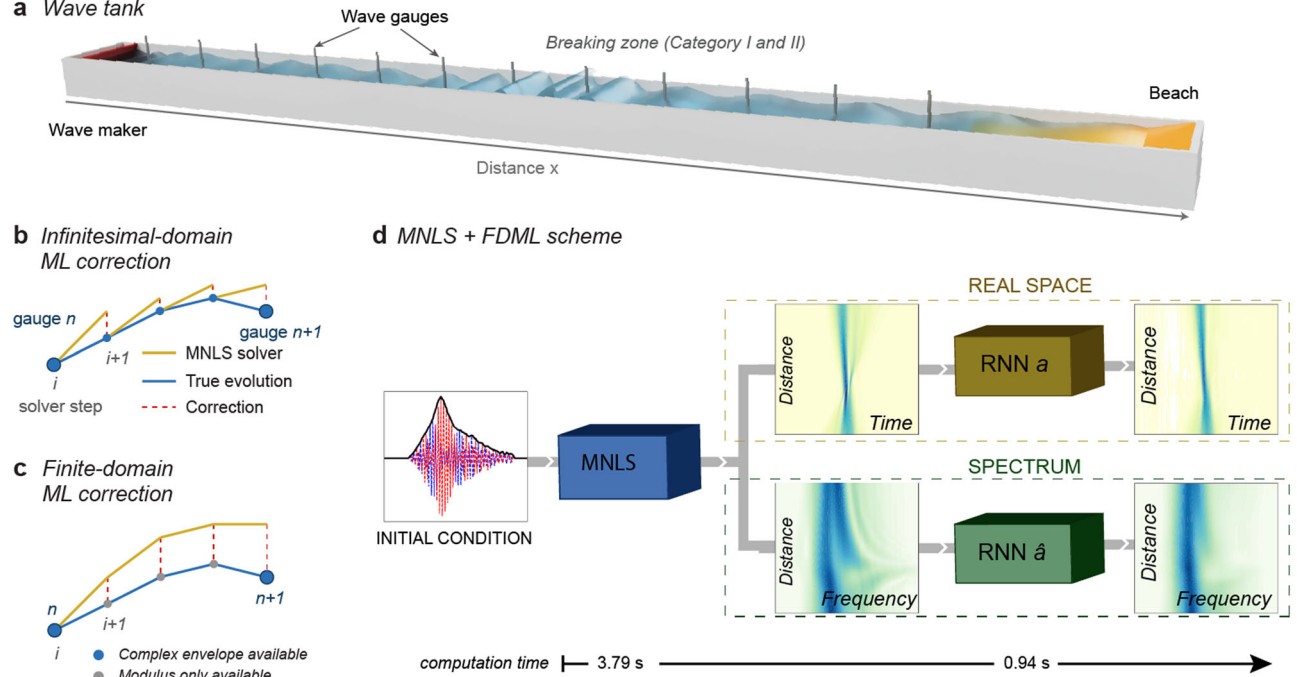

**Fig. 7 Overview of methods. a** Schematic drawing of the wave tank. For waves Categories I and II experiment the wave tank was 40 m long, 2.7 m wide, with a water depth of 0.80 m. A total of 12 wave gauges were placed at ~3 m distances along the center-line of the tank, starting 3.7 m from the wavemaker (red). For wave Category III, the wave tank was 30 m long, 1 m wide, with a water depth of 0.70 m. A total of 24 wave gauge positions were placed 0.83 m apart, starting at 3.0 m from the wavemaker. **b** For the infinitesimal-domain method, access is needed to the complex envelope at each solver step, to serve as the initial condition for the MNLS. **c** For the finite-domain method the complex envelope is only needed at the wave gauge positions, and only the absolute value of the envelope is needed at each solver step. **d** When given a (complex) initial condition, the MNLS solver can predict the evolution for an arbitrary propagation distance. The two separate RNNs then independently provide corrections to the spectral evolution and the physical-space evolution, respectively.

given by:

$$b_0 = 1 - b_F, \quad b_{-1} = (1 - b_0 + \alpha)/2, \quad b_{+1} = (1 - b_0 - \alpha)/2, \quad (4)$$

where $b_F$ is sideband fraction, and $\alpha$ the sideband unbalance. The dimensional signal for the wavemaker can be obtained as $\eta(0, t) = \tilde{a}_0 a(0, \tau) e^{i\omega_0 t}$. Due to modulation instability (MI)[39,81], the spectrum will broaden, and the amplitude of the wave will increase, attaining a maximum at the focus position $\xi_f$ after which the modulation will decrease: the FPUT recurrence cycle[50–52]. By selecting the initial steepness $\tilde{a}_0 k_0 / \sqrt{2}$, $b_F$, $\alpha$, and $\Omega_M$, we can set the steepness at $\xi_f$.

To generate a variety of waves, parameters were drawn from the following ranges: $f_0 = \omega_0/(2\pi) \in [1.30, 1.55]$ Hz, $\tilde{a}_0 k_0 \in [0.12, 0.25]$, $b_F \in [0.01, 0.1]$, $\Omega_M \in [0.7, 1.55]$, $\psi \in [0, 2\pi]$ and $\alpha \in [-0.16, 0.16]$. A data set of 258 wave trajectories were generated and measured. Figure 1 shows the evolution of a breaking wave with initial parameters $f_0 = 1.40$ Hz, $\tilde{a}_0 k_0 = 0.19$, $b_F = 0.09$, $\Omega_M = 1$, $\psi = \pi$, and $\alpha = 0$.

**Wave Category II**. For the focused irregular waves, the JONSWAP spectral density $S(\omega)$ is characterized by the shape parameter $\gamma$:

$$S(0, \omega) = \frac{Kg^2}{\omega^5} \exp\left[-\frac{5}{4}\left(\frac{\omega_p}{\omega}\right)^4\right] \gamma^r, \quad (5)$$

with $r = \exp\left[-\frac{(\omega - \omega_p)^2}{2\sigma_s^2 \omega_p^2}\right]$. Here, $\omega_p$ is the peak frequency in rad/s and $\sigma_s$ the spectral width. $\sigma_s = 0.07$ for $\omega \leq \omega_p$ and $\sigma_s = 0.09$ for $\omega > \omega_p$. As the parameter $K$ scales the entire spectrum, it can be chosen to obtain the appropriate significant wave height $H_s$. When mimicking a random sea state, the phases of the spectrum are randomized. To focus the spectrum, the phases are chosen such that components $a_n(\omega_n)$ of the JONSWAP spectrum superimpose to a maximum at the same position in the tank, $x_f$, creating a maximum amplitude of the wave group at this position, and possibly a breaking event. The surface elevation at the wavemaker reads:

$$\eta(0, t) = \sum_n a_n \cos(\omega_n t + k_n x_f). \quad (6)$$

Parameters were drawn from the following ranges: $\gamma \in [2, 5]$, $H_s \in [10, 60]$ mm, $f_p = \omega_p/(2\pi) \in [0.5, 1.25]$ Hz, $x_f \in [10, 24]$ m, such that each initial condition was generated based on a unique combination of these parameters. A data set of 180 wave trajectories were generated and measured. Note that for both wave categories, parameters cannot be selected randomly, as not every combination leads to a

breaking event within the length of the tank. Figures 2 to 4 show the evolution of a breaking wave with initial parameters $f_p = 0.91$ Hz, ($\omega_p = 5.7$ rad/s), $\gamma = 4$, $H_s = 30$ mm and $x_f = 14$ m.

**Wave Category III**. For the random irregular waves, the JONSWAP spectral density eq. (5) is used. Instead of manipulating the phases to exhibit dispersive wave focusing, they are randomly drawn from a uniform distribution for each wave frequency in the spectrum. Parameters for the initial condition were $\gamma = 3.3$, $f_p = 1.25$ Hz, and four different significant wave height values: $H_s = 25, 34, 41, 54$ mm. For each wave height, three 20-min experiments (about 1500 wave periods) were conducted. Shorter experiments were performed to create a test set with properties: $\gamma = 6.0$, $f_p = 1.25$ Hz, $\sigma_s = 1$, $H_s = 44$ mm.

**Neural network and algorithm**. The goal of training is to teach the network the discrepancy between the MNLS propagation prediction and the true propagation from measurements over an arbitrary number of solver propagation steps when started from the same initial condition. As we do not have the phase information that allows us to Fourier transform the envelope between the time and frequency domain at each solver step, we correct the evolution of the time and frequency domain separately, by using two separate Neural Networks with identical architecture. One in the time domain and the other in the frequency domain. Below, we outline the structure of the algorithm for Wave Categories I and II (see SI section 6 for further details). The procedure for Wave Category III is slightly different, as this involved segmenting a continuous time series and is outlined in SI section 5.1.

- **Obtaining the ground truth at the solver step** —The wave tank experiments need to be interpolated such that the ground truth is available at very (fixed-size) solver propagation step $\Delta\xi$ instead of only at the wave gauge positions. The Nyquist–Shannon sampling theorem provides a lower bound for the sampling frequency to capture all the information from a continuous signal of final bandwidth. Following this, the slowly varying envelope is sufficiently sampled by our 12 gauges in space, and by the wave gauge sampling frequency in time. As we can only interpret the modulus of the envelope of the measurement and not its phase, we perform a spline interpolation on the modulus of the wave tank measurements separately in the time and frequency domains. Hereby moving from the envelope at discrete locations of the wave gauges $a_{\text{true}}(\xi_{\text{wg}})$ to the modulus of the

envelope-field at interval $\Delta\xi$ for both domains:

$$a_{\text{true}}(\xi_{\text{wg}}) \longrightarrow |a|_{\text{true}}(\xi), |\hat{a}|_{\text{true}}(\xi) \qquad (7)$$

- **Data augmentation**—Simply using the entire experiment would not yield enough data, and we, therefore, augment the data by creating smaller propagation segments. In order to account for different stages of evolution of the wave, we can choose training samples starting at different positions $\xi_0$. Note that this is different from simply cutting the entire propagation comparison of MNLS and experiment into smaller pieces, as each segment requires its own MNLS simulation starting from its own initial condition. For each experiment, three segments lengths are selected. The starting position can only be at wave gauge positions $\xi_k$, as these are the only locations with access to the complex field (modulus and phase). An MNLS simulation can be started from this initial condition $a(\xi_k, \tau)$, a segment length $n_{\text{steps}}$, yielding $a_{\text{MNLS}}(\xi_k, \xi_k + n_{\text{steps}}\Delta\xi, \tau)$. This prediction can then be compared to the same stretch of the measured field, or true evolution $a_{\text{true}}$ in eq. (7). Since only the modulus of the envelope is available at the solver step, an input-output training pair $s$ is the modulus of the envelope

$$s = [\text{input, output}] = \left[ |a|_{\text{MNLS}}(\xi_k, \xi_k + n_{\text{steps}}\Delta\xi), |a|_{\text{true}}(\xi_k, \xi_k + n_{\text{steps}}\Delta\xi) \right]. \qquad (8)$$

Similarly, a separate training pair $\hat{s}$ can be created for the modulus of the spectrum of the envelope. To limit the number of free parameters, the number of modes (and, correspondingly, the length of the time-vector) is truncated to $n_t = 512$. The input and output therefore each have matrix dimensions $[n_{\text{steps}}, n_t = 512]$. By varying the segment length and the starting gauge, many pairs $s \in S$ can be created. Note that the network is never trained on the entire propagation length available in the tank.

- **Neural Network**—A neural network is a nonlinear function $F_{\text{NN}}$ of its parameters $\beta$ (the weights and biases) and the input:

$$|a|_{\text{pred}} = F_{\text{NN}}(\beta, |a|_{\text{MNLS}}). \qquad (9)$$

The goal is to optimize $\beta$ such that the mean squared error (MSE) of the prediction and the true value of the envelope is minimized. We employ a separate network for the time and frequency domain. For the time-domain RNN, this cost function is simply defined as:

$$J(\beta) = \frac{1}{N} \sum_{i=1}^{N} (F_{\text{NN}}(\beta, |a|_{\text{MNLS}})_i - |a|_{\text{true},i})^2 = \frac{1}{N} \sum_{i=1}^{N} (|a|_{\text{pred},i} - |a|_{\text{true},i})^2, \qquad (10)$$

where $i$ the grid point, and $N = n_{\text{steps}} \times n_t$, the total number of grid points. The definition for the frequency-domain network is the counterpart of eq. (10) for $|\hat{a}|$. To find the network parameters $\beta$ that minimize the cost function, $J(\beta)$, the *Adam* stochastic gradient descent method with bounded gradient is used. Details of the optimization include:

For this optimization problem, different network architectures can be chosen for $F_{\text{NN}}$. Our work employs an RNN, for which the weights of the hidden states are updated based on the previous propagation step and passed on to the next recursively. As such, RNNs are able to return an output sequence, incorporating the memory of all the propagation steps in the input sequence. This is crucial to detect breaking signatures along the evolution. The main property of the finite-domain correction is that the system evolves for several propagation steps, after which the correction is applied. As a consequence, the correction uses information from future propagation steps and is non-causal. To mediate the problems of vanishing and exploding gradients that a fully-connected RNN would have, we employ the long short-term memory (LSTM) method[82].

The measurement data are divided into a training (80%), validation (15%), and test (5%) set. The first is used to minimize eq. (10), the second to evaluate performance after each training epoch (a complete pass through the training set), and the latter is never used in the optimization process, and only serves to validate the performance of the model.

Since wave breaking does not possess any conserved quantities for the envelope evolution, no physical constraints can be added as either soft or hard constraints.

As the time-vector is of length 512, consequently, the input and output layers must consist of 512 units. The LSTM layer is connected to the input and consists of 128 hidden units, followed by one dense, or fully-connected, layer with 64 units. To promote generality, the dense layer has a dropout of 0.1, meaning this fraction of neurons will be randomly turned off during training. The dense layer is connected to the output layer. All layers have a leaky rectified linear unit (ReLu) activation function. The resulting model has a total of 369,728 parameters (degrees of freedom). Note that the architecture of the LSTM layer stays the same for all propagation steps; more propagation steps do not add more degrees of freedom.

Note that for prediction, the MNLS trajectory must have the same time and propagation step as during training: $\Delta\xi = \Delta\xi_{\text{train}}$. Similarly for the time step

$\Delta\tau = \Delta\tau_{\text{train}}$. Finally, the network is trained on data normalized by the maximum value of the initial condition.

The MNLS solver has periodic boundary conditions, therefore it is equivariant to translations of the input in $\tau$. To account for this in physical space we add 40 random translations $n_{\text{tr}} \in [0, n_t = 512]$ of the input and output vectors. Note that in frequency space, these shifts do not affect $\hat{a}$. Additional examples of these translated results are included in SI section 2.2.

- **MNLS + FDML correction**—Once the network is trained, MNLS + FDML model can be utilized as illustrated in figure Fig. 7d. While the network has been trained on shorter segments, it can be used for longer, or arbitrary propagation length, and on unseen data. Like the MNLS, an initial value or deterministic wave-forecasting problem is solved. That is, for a given initial condition at a position $x_0$, one wants to know the evolution up to and including a point $x_1$. If the initial condition is a time-series measurement of the surface elevation, the complex envelope $a(x_0, t)$ has to be obtained using the Hilbert transform. Subsequently, the MNLS solver can be used to propagate the solution forward to $x_1$. Over this finite stretch of evolution, the RNN applies a correction to the modulus of the time-domain envelope amplitude, or on the frequency-domain envelope amplitude to correct for breaking effects. Figures 1 and 2 show the entire propagation length available in the tank merely because this offers the most extensive comparison.

## Data availability

The wave tank measurement data generated in this study have been deposited in the Zenodo database under accession code https://zenodo.org/record/6326470.

## Code availability

The code used in this study has been deposited in the Zenodo database under accession code https://zenodo.org/record/6338618.

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

## Acknowledgements

D.E. acknowledges financial support from the Swiss National Science Foundation (Mobility Fellowship P2GEP2-191480). T.P.S. and D.E. acknowledge the ONR Grants N00014-21-1-2357 and N00014-20-1-2366. J.K. and D.E. acknowledge SNSF 200020-175697. Y.H. was supported by the University of Sydney International Scholarship (USydIS). T.S.vd.B. was supported by a Royal Academy of Engineering Research Fellowship. The authors acknowledge Thomas Adcock and Ye Li for access to the external data and thank Maura Brunetti and Alexis Gomel for fruitful discussions and Michel Moret for providing access to the computation infrastructure.

## Author contributions

D.E. and T.P.S. designed the research; D.E. H.B., A.C., Y.H., J.K., and C.L. performed experiments; D.E., Y.H., A.C., J.K., T.P.S., and T.S.vd.B. analyzed the data; D.E. and T.P.S. developed the FDML framework. D.E., T.P.S., J.K., and T.S.vd.B. wrote the paper.

## Competing interests

The authors declare no competing interests.
