## [Peer Review File · Nature Communications]

REVIEWER COMMENTS

Reviewer #1 (Remarks to the Author):

Key results:

The authors present a blended model for deep water waves using the modified nonlinear Schrodinger (MNLS) equation for wave evolution and a finite domain machine learning (FDML) method to predict and account for breaking. This coupled MNLS + FDML model compares well with measurements from lab tank experiments for unidirectional perturbed plane waves and unidirectional focused irregular (JONSWAP spectrum) waves, the latter providing more realistic conditions. This work is a foundational step towards combining a physical wave model and data-driven machine learning to explicitly and implicitly capture enough of the physics of wave breaking to predict deep water wave evolution efficiently and accurately. This work tests a narrow range of conditions but provides a basis upon which more complex physics and wave type models can be developed.

Validity:

Assuming that these example cases are representative, the authors' conclusions are reasonable and substantiated. There were 258 plane wave experiments and 180 irregular focused wave experiments run, but only a few example cases are shown in the manuscript and supporting information. I do think the example cases are an appropriate and understandable way to demonstrate the method and describe the results. However, I think these examples could be strengthened by the inclusion of some statistics on overall performance for each wave type (even if only included in supplementary information) to support the use of these examples as representative of the full suite of results. For example, could you compute difference or percent difference between the MSE (as a function of distance along the tank, just like in figures 1 and 2 panels d and h) for MNLS and MNLS + FDML and show those curves for all experiments as well as their ensemble average for each wave type in time and frequency space?

Significance:

This work aims to overcome the shortcomings of direct numerical simulation and spectral models to achieve a practical wave-resolving forecasting in deep water. Direct numerical simulation is computationally expensive (in 3D and 2D) which limits the space and time over which predictions can be made. Spectral models rely on empirically derived dissipation parameterizations of breaking and do not resolve individual waves. The MNLS + FDML method solves the computationally expensive problem by using simplified physical equations to describe non-breaking wave evolution and machine learning techniques to predict and account for breaking in the model. Additionally, the finite domain method, used here out of necessity, might also be helpful for future field tests because field data for training the neural network could be sparse like the lab flume data (though remote sensing measurements of the real-world wave field may provide ample density of training data, see comments in Data and methodology). The authors also clearly explain the limitations of this new method, which enables the community to appropriately utilize and build upon this work.

This work is a significant first step to using machine learning in conjunction with wave physics to predict real-world wave fields. However, only unidirectional narrow-banded deep-water waves were tested. The authors explain that the method could be applied to any given wave type, so long as the RNN is trained on each new wave type. What does this imply for a mixed sea state in real life? Is this method flexible enough to model a real-world wave field? The machine learning methodology is incredibly powerful, but if the training data is sparse and the wave field is complicated, the machine learning skill could be compromised, as acknowledged by the authors in their discussion of the finite domain

technique. I am excited to see the next steps proposed by the authors, extending to different wave types and an infinitesimal-domain blended model to predict real-world waves.

Data and methodology:

I have a few basic questions about the decisions the authors made for their MNLS equation. I am not questioning the choices, but I think readers might benefit from a bit more explicit groundwork before the discussion section. For example, the authors reference their previous work in Armaroli et al. (2018) [37] that included wind forcing in the MNLS, but they do not include that term here. Adding wind forcing would make the model more like the real world, but the experimental waves were not wind-generated so this term is not appropriate to include. In lines 174-178, the authors mention the Trulsen and Dysthe (1996) [59] MNLS equation with modifications for broader bandwidth waves, which would also mimic real world waves better. However, the experimental waves were unidirectional and narrow-banded, so the author's choice of equation fits the experimental design they were trying to model.

In the Introduction, could the authors add a brief explanation of the types of forcing that could be included in their MNLS and why they chose to retain certain terms and not others? I think this would be helpful for when readers get to the Discussion and the implications of these options are mentioned. I am mostly interested in this in the context of understanding the applicability of the model and next steps that would be needed to scale towards real-world waves (relative importance of viscous damping, wind forcing, modifications for broad-band waves).

The details of the RNN are beyond the scope of my expertise, as noted in section Expertise. However, I have some comments about data requirements that may or may not be within the scope of this paper. The authors explain that the limitation to using the more complicated model for directionally spread seas is getting sufficiently resolved ground truth 2D wave field data, and potentially switching from finite domain to infinitesimal domain methods. Could the authors provide a general order of magnitude estimate of the resolution of data required to incorporate the modifications to the MNLS for broad-banded waves? Or provide examples of the types of measurements that would be sufficient? For instance, would a 3D reconstruction of the wave field from ship-based (or deployed in the lab) stereo imagery (Schwendeman and Thomson, 2017) be sufficient for RNN training on directionally spread seas?

Analytical approach:

The analytical approach is well-documented in the text and supplementary information. As suggested in Validity above, I do think the results presented would benefit from the inclusion of bulk analysis and statistics on overall performance for each wave type.

Suggested improvements:

See suggestions in other sections.

Clarity and context:

These are all minor points to improve clarity and context.

- I think it is important to include in the abstract that this work applies to deep-water waves and breaking. I understand that the NLS equation applies only to deep-water waves, but since NLS doesn't get mentioned until the body of the text, it would be helpful to the

reader to know the scope of the paper up front.

- In the early figures where the breaking position is noted, I suggest moving the annotation either to the left side of the figure or within the figure along the dotted line. Currently, it looks like the breaking position references a specific color on the amplitude color bar, especially in figures 1 and 2, which could be confusing for readers.

- "PDE" is never spelled out. I know it refers to partial differential equations (PDE), but please include this at its first mention in the text.

References:

References to previous relevant literature are appropriately used throughout the text.

Expertise:

I understand the basics of machine learning and neural networks, but the specific details of the recurrent neural network and long short-term memory methods are outside the scope of my expertise. I communicated this to the editor, who ensured another reviewer would be qualified to comment on those aspects of the paper.

References included in this review:

Schwendeman, M. S., & Thomson, J. (2017). Sharp-crested breaking surface waves observed from a ship-based stereo video system. *Journal of Physical Oceanography*, 47(4), 775-792.

Thank you for your contribution to our scientific community,
Roxanne J Carini, PhD
Senior Oceanographer, NANOOS
Applied Physics Laboratory,
University of Washington

Reviewer #2 (Remarks to the Author):

Review of the paper
“Nonlinear wave evolution with data-driven breaking”
by D. Eeltink et al.

This paper investigates the numerical propagation of nonlinear water waves and intends to address the complex configuration in which breaking events appears during the course of the wave evolution. The interesting idea used in the present work is to combine a classical non-breaking propagation scheme (here based on MNLS) with machine learning to account for the effects of wave breaking. The training data are based on experiments in water tanks. This original work is of great interest to the nonlinear water wave community since the accurate and efficient description of breaking sea states is still an open question. The application of the methodology developed can for sure be extended to other fields of physics. Then, I would say that the work fits the scope of Nature Communications.

However, the paper is in my opinion not suitable for publication in its present form. My main concern is related to the lack of generality of the results presented. The training data sets are specific to the cases tested (namely Wave Category I and Wave Category II) and it is consequently not obvious that the present methodology is usable in a practical configuration in which one does not know what kind of breaking will appear in the sea state. A so-called external data is presented to partly address this issue but even if the scale is different as well as the spectrum shape, it is still a focused event. Dynamics of a focused wave will be very similar and even if it brings new insight, it is for me not sufficient to be sure of its possible generalization to any kind of sea states.

The major points I can indicate that may help the authors to improve their manuscript (and detail my previous statement):

- Looking at the training data sets characteristics and the one tested in Sec. 2.1 and 2.2, the simulated waves are probably very close to one original data set. Then, even if it is an interesting preliminary step to test the procedure, additional test cases are necessary to me.
- The lack of generality of the work is partly pointed out by the authors in Discussion section. However, they point toward another methodology based on infinitesimal-domain method to possibly solve the issue. Then, what are the salient point one can keep from this work if it needs to be modified for practical application?
- Methods - I am not sure to have understood in details the two RNNs that are referred to in the text and in Figure 6. Does the resolution occur in the real space or in the spectral space? Or are both the resolution done at the same time? If so, how is ensured the compatibility between both?
- l.197: it is explained that bound modes are filtered from the measurement signal: how is it done practically for a broad-banded spectrum such as JONSWAP?
- Appendix B.1: How is it possible that MNLS is working worse than MNLS-FDML in a non-breaking wave condition? I think it needs to be discussed briefly.

- Appendix C.1: it is said that FDML applies over-damping in this non-breaking condition. I think this is an issue toward the possible generalization of the use of the method. Is it possible to enhance this with additional training data (non-breaking)? Or do you expect that those additional data will alter the results presented in breaking conditions?

Reviewer #3 (Remarks to the Author):

Nature Communications, Manuscript Number NCOMMS-21-30443

The authors present the very interesting problem of predicting wave breaking at high resolution, at a low computational cost, using hybrid physics and machine learning (ML) techniques. A noteworthy aspect is that the authors have used experimental data instead of numerical simulation data that is traditionally used in such studies. This adds to the complication since the experiments provide data only at specific spatial locations with gauges, as opposed to the entire field (which numerical simulations can, but are very expensive).

As a solution, the authors propose a hybrid ML + physics approach that they claim exploits both the Modified nonlinear Schrodinger (MNLS) equation while using a neural network to "correct" it with experimental data. While the motivation and overall direction is outlined well, I unfortunately had a very difficult time a) Extracting the actual *details* of the method that is proposed, and b) In seeing the novelty in this work. Figure 6 is helpful, but not enough for the detailed mathematical content for a Nature publication, as it is. In Pg. 2, Lines 68-70 the authors say "...applies an RNN". However, there is no exact information on what makes "FDML" any different from just feeding data to an RNN. In fact, there are plenty of results shown, but no information about how "MNLS + FDML" model is developed! In fact, the only relevant section is the one on LSTM and Eqn. (7), which makes it very confusing to understand exactly how the MNLS "interfaces" to the LSTM and why it is a novel enough approach to warrant publication in Nature Comms, since there are several papers that have already demonstrated some flavor of this (Too many to cite here). While I can see an innovation being on how they use experimental as opposed to simulation data, this aspect alone is not enough to carry the whole paper, since the rest of the method is not explained clearly. For this reason, *I am forced to not recommend publication of this manuscript in its Current form*. However, I highly encourage the authors to resubmit so the algorithm and innovation is clearly outlined, with the appropriate citations to relevant, related works already in this field.

Some specific questions:

- 1) Where do $|a_{MNLS}|$ and $|a|_{true}$ come from? Lines 239 and 240 give some indication, but this is a very important part of the authors' contribution, and must be explicitly defined in an equation and/or figure for clarity. From the information provided in the manuscript, I fail to understand the novelty in ML, when the approach just uses a standard LSTM. I would *guess* that the innovation is somewhere in how the MNLS and LSTM are interlinked, but I cannot find that in the paper.
- 2) The authors make no comment on the stability or boundedness of the trajectory predicted by the LSTM, apart from the spline interpolation performed on the gauge data. While 12 gauges are enough for this case according to the authors, it provides no scientific basis that this method is generalizable beyond this specific experimental dataset. Additionally, LSTMs have no implicit boundedness guarantees, so there is little reason for the situation in Fig 6(c) to hold true.
- 3) How do you ensure that physical constraints are obeyed? The LSTM loss functions are "soft" constraints as they merely act as a suggestion to the optimizer, but do not guarantee any fidelity to constraints - There are papers on "hard" constraints that do this job and are much more robust and accurate than soft constrained neural networks (I request the authors to read those papers - there are many and I don't want to bias by providing my preferred references) , and these approaches are the current state of the art!

Finally, In my opinion, the goal of a research manuscript is to provide enough detail for someone to implement their technique from scratch. While making the code available is a

commendable move, it is not a substitute for the actual text. There is indeed a possibility that there indeed is something very unique in the MNLS + FDML algorithm that I might have missed due to lack of relevant details from the authors, but I cannot see it from the content in the paper. However, readability is very important to readers who may not be from a specific field (wave breaking or machine learning), and is severely lacking for those outside this specific niche. That said, I would be open to reassessing the paper on its technical merits once the requisite details are added.

Reviewer #1 (Remarks to the Author):

We would like to thank the reviewer for their positive response and helpful detailed comments, which we address individually below. Our response to the reviewer's report reproduced below is in italics and indented.

Key results:

The authors present a blended model for deep water waves using the modified nonlinear Schrodinger (MNLS) equation for wave evolution and a finite domain machine learning (FDML) method to predict and account for breaking. This coupled MNLS + FDML model compares well with measurements from lab tank experiments for unidirectional perturbed plane waves and unidirectional focused irregular (JONSWAP spectrum) waves, the latter providing more realistic conditions. This work is a foundational step towards combining a physical wave model and data-driven machine learning to explicitly and implicitly capture enough of the physics of wave breaking to predict deep water wave evolution efficiently and accurately. This work tests a narrow range of conditions but provides a basis upon which more complex physics and wave type models can be developed.

Validity:

Assuming that these example cases are representative, the authors' conclusions are reasonable and substantiated. There were 258 plane wave experiments and 180 irregular focused wave experiments run, but only a few example cases are shown in the manuscript and supporting information. I do think the example cases are an appropriate and understandable way to demonstrate the method and describe the results. However, I think these examples could be strengthened by the inclusion of some statistics on overall performance for each wave type (even if only included in supplementary information) to support the use of these examples as representative of the full suite of results. For example, could you compute difference or percent difference between the MSE (as a function of distance along the tank, just like in figures 1 and 2 panels d and h) for MNLS and MNLS + FDML and show those curves for all experiments as well as their ensemble average for each wave type in time and frequency space?

We thank the reviewer for their suggestion and have included the proposed statistics on overall performance of the model as a new figure (Figure 5). For completeness, we note that while a large number of experiments was available for our study, 80% was used as training data and 15% as validation data, used to check its optimization. Showing statistics on the model performance on these subsets is commonly not done as this does not attest to the model's performance on unseen data. This leaves 5% of the experiments to be used as test data never seen by the network during optimization/training, which is used in Figure 5. Note that for training, the data is augmented, to different starting points and segment lengths (see Methods), effectively creating many more training samples from one experimental run. However, the test cases we plot are only from one starting point -the first wave gauge-, and for the entire length of the tank. We have also added a tiled overview of these test cases in Supplementary Information Sections 2 and 3.

Significance:

This work aims to overcome the shortcomings of direct numerical simulation and spectral models to achieve a practical wave-resolving forecasting in deep water. Direct numerical simulation is computationally expensive (in 3D and 2D) which limits the space and time over which predictions can be made. Spectral models rely on empirically derived dissipation parameterizations of breaking and do not resolve individual waves. The MNLS + FDML method solves the computationally expensive problem by using simplified physical equations to describe non-breaking wave evolution and machine learning techniques to predict and account for breaking in the model. Additionally, the finite domain method, used here out of necessity, might also be helpful for future field tests because field data for training the neural network could be sparse like the lab flume data (though remote sensing measurements of the real-world wave field may provide ample density of training data, see comments in Data and methodology). The authors also clearly explain the limitations of this new method, which enables the community to appropriately utilize and build upon this work.

This work is a significant first step to using machine learning in conjunction with wave physics to predict real-world wave fields. However, only unidirectional narrow-banded deep-water waves were tested. The authors explain that the method could be applied to any given wave type, so long as the RNN is trained on each new wave type. What does this imply for a mixed sea state in real life? Is this method flexible enough to model a real-world wave field? The machine learning methodology is incredibly powerful, but if the training data is sparse and the wave field is complicated, the machine learning skill could be compromised, as acknowledged by the authors in their discussion of the finite domain technique. I am excited to see the next steps proposed by the authors, extending to different wave types and an infinitesimal-domain blended model to predict real-world waves.

We thank the reviewer for their positive evaluation and recognizing the significance of our work. The reviewer raises an important question on how to move forward to a real-world sea state, which have now included in the Discussion of the manuscript and the Results section Wave Category III: random irregular waves.

Regarding (random) real-world wave fields, two main difficulties arise: First, one has to take into account that for a finite measurement at point in space, over a time $t[t_1, t_2]$ due to the longest waves moving faster than the slowest waves, the predictable region narrows (see e.g. figure 1 of Galvagno (2021)), yielding a finite prediction horizon also for non-breaking waves. Secondly, in the deterministic sea states, the effect of breaking has clearly distinguishable signatures in the evolution, as detailed in the manuscript. For random sea states, the breaking signatures are much less clear.

Following the reviewer's question, we have performed new experiments to add a third wave category to the manuscript: unidirectional irregular waves. That is, a JONSWAP spectrum with random phases. As the effect of wave breaking on the spectrum is not easily defined, we compare experiment and prediction only in the time domain. Here, the MNLS predicts unphysical high peaks where in reality the wave would break. Instead of taking repeated short measurements as for Wave Categories I and II, we have one long time series that is then sectioned. Apart from that, the same method as for the other wave categories can be applied. This demonstrates that our method can indeed be readily extended to an irregular sea state. We do remark that naturally, the comparison of the

envelope is less informative than the surface elevation directly. As detailed below, the exact same setup can be used with a different physics-based model, measurement combination.

To tackle even more realistic sea states, we anticipate two future directions of development, distinguishing those that make use of the infinite-domain and those that make use of the finite-domain approach. In the finite-domain approach, the same approach as in this paper can readily be used for other combinations of models and measurement techniques depending on the purpose of the study, such as deterministic wave forecasting, statistics, or extreme event detection. For each purpose the blended approach as in this work can be applied: a model that captures the basic physics without breaking, a machine learning layer with memory, and measurements that track the evolution at finite intervals. The challenge, especially in two-dimensional sea states, will be to obtain sufficient experimental data. We discuss this challenge in more detail in the reviewer's question on Data and Methodology.

Data and methodology:

I have a few basic questions about the decisions the authors made for their MNLS equation. I am not questioning the choices, but I think readers might benefit from a bit more explicit groundwork before the discussion section. For example, the authors reference their previous work in Armaroli et al. (2018) [37] that included wind forcing in the MNLS, but they do not include that term here. Adding wind forcing would make the model more like the real world, but the experimental waves were not wind-generated so this term is not appropriate to include. In lines 174-178, the authors mention the Trulsen and Dysthe (1996) [59] MNLS equation with modifications for broader bandwidth waves, which would also mimic real world waves better. However, the experimental waves were unidirectional and narrow-banded, so the author's choice of equation fits the experimental design they were trying to model.

In the Introduction, could the authors add a brief explanation of the types of forcing that could be included in their MNLS and why they chose to retain certain terms and not others? I think this would be helpful for when readers get to the Discussion and the implications of these options are mentioned. I am mostly interested in this in the context of understanding the applicability of the model and next steps that would be needed to scale towards real-world waves (relative importance of viscous damping, wind forcing, modifications for broad-band waves).

We have now better framed what our MNLS does and does not model to provide context for the rest of the paper aimed at a broad readership. We have added a clarification in the introduction of mechanisms that can influence the propagation of the wave-field, and have explained what the MNLS version we present does and does not model. Specifically, we explain that the broad-band modification of Trulsen and Dysthe (1996) only increases accuracy for the linear dispersion relation in time. One can see this by looking at the dispersion relation in time: ($\omega = \sqrt{gk}$), which can be expanded up to an arbitrary order. However, the dispersion relation in space ($k = \sqrt{\omega^2/g}$) is finite due to the quadratic term, and therefore expanding it does not add any terms to the unidirectional spatial evolution (see Trulsen et al. (2000)). In addition, we explain that that we do not consider wind-forcing, to disentangle the effect of the different processes on the wave field. That being said, we agree with the reviewer that wind-forcing and subsequent wave

breaking will be an interesting dynamical interaction to study. We have added this to the Discussion section.

The details of the RNN are beyond the scope of my expertise, as noted in section Expertise. However, I have some comments about data requirements that may or may not be within the scope of this paper. The authors explain that the limitation to using the more complicated model for directionally spread seas is getting sufficiently resolved ground truth 2D wave field data, and potentially switching from finite domain to infinitesimal domain methods. Could the authors provide a general order of magnitude estimate of the resolution of data required to incorporate the modifications to the MNLS for broad-banded waves? Or provide examples of the types of measurements that would be sufficient? For instance, would a 3D reconstruction of the wave field from ship-based (or deployed in the lab) stereo imagery (Schwendeman and Thomson, 2017) be sufficient for RNN training on directionally spread seas?

We thank the reviewer for this very useful reference that we were not aware of. We have included this suggestion in the discussion. We have given the extension to more dimensions more thought thanks to this suggestion, and have adjusted our discussion accordingly.

One issue for ship-based measurements is of course that other real-world processes such as winds cannot be excluded. As the reviewer suggests, lab measurements can alleviate this issue. Often, 2D simulations that evolve in time (i.e., $\eta(x,y,t)$) assume a periodic domain in space, or require reflections from the sidewalls as in a numerical wavetank, adding another complication. Therefore, comparing data and simulation -even for non-breaking waves- that evolve in time would require a sufficiently large spatial domain, such that these boundaries do not matter, or reflections can be accounted for. To our knowledge, it would be difficult to measure such a large domain with stereo imaging. A number of the authors of the paper are involved in a large-scale project that sets out to obtain such data with an array of gauges instead.

To generalize our approach to directional sea states, the 1D MNLS equation will need to be replaced by the 2D MNLS or a model that directly describes the surface elevation (instead of the envelope) and is less restrictive on bandwidth, such as higher-order spectral methods (HOSM) or the Zakharov Equation.

In specific, the reviewer asks to provide a general order of magnitude estimate of the resolution of data required to incorporate the modifications to the MNLS for broad-banded waves:

The Nyquist–Shannon sampling theorem provides a lower bound for the sampling frequency such that it permits capture of all the information from a continuous signal of final bandwidth. The sampling frequency must be twice that of the highest frequency in the continuous signal.

If the stereo-imaging technique would be used in a blended method with a model for the surface elevation instead of the envelope, and one could resolve the 0th harmonic of the surface elevation in the lab, a spatial resolution $\lambda_0/4$ would be needed. In Schwendeman and Thomson, 2017 a grid spacing of 25 cm is mentioned. This should be sufficient, for waves typically generated in larger wave tanks (λ_0 order of 2-5m).

Analytical approach:

The analytical approach is well-documented in the text and supplementary information. As suggested in Validity above, I do think the results presented would benefit from the inclusion of bulk analysis and statistics on overall performance for each wave type.

We refer to our reply in the Validity section.

Suggested improvements:

See suggestions in other sections.

Clarity and context:

These are all minor points to improve clarity and context.

- I think it is important to include in the abstract that this work applies to deep-water waves and breaking. I understand that the NLS equation applies only to deep-water waves, but since NLS doesn't get mentioned until the body of the text, it would be helpful to the reader to know the scope of the paper up front.

We now mention this in the abstract.

- In the early figures where the breaking position is noted, I suggest moving the annotation either to the left side of the figure or within the figure along the dotted line. Currently, it looks like the breaking position references a specific color on the amplitude color bar, especially in figures 1 and 2, which could be confusing for readers.

We thank the reviewer for pointing this out; we have adapted the figures

- "PDE" is never spelled out. I know it refers to partial differential equations (PDE), but please include this at its first mention in the text.

We have now included the definition the first time we mention partial differential equations.

References:

References to previous relevant literature are appropriately used throughout the text.

Expertise:

I understand the basics of machine learning and neural networks, but the specific details of the recurrent neural network and long short-term memory methods are outside the scope of my expertise. I communicated this to the editor, who ensured another reviewer would be qualified to comment on those aspects of the paper.

References included in this review:

Schwendeman, M. S., & Thomson, J. (2017). Sharp-crested breaking surface waves observed from a ship-based stereo video system. *Journal of Physical Oceanography*, 47(4), 775-792.

Thank you for your contribution to our scientific community,

Roxanne J Carini, PhD

Senior Oceanographer, NANOOS

Applied Physics Laboratory,
University of Washington

References used in this reply:

Galvagno, M., Eeltink, D., & Stuhlmeier, R. (2021). Spatial deterministic wave forecasting for nonlinear sea-states. Physics of Fluids, 33(10), 102116.

Trulsen, K., Kliakhandler, I., Dysthe, K. B., & Velarde, M. G. (2000). On weakly nonlinear modulation of waves on deep water. Physics of Fluids, 12(10), 2432–2437.

Reviewer 2:

We would like to thank the reviewer for their positive response and helpful detailed comments, which we address individually below. Our response to the reviewer's report reproduced below is in italics and indented.

This paper investigates the numerical propagation of nonlinear water waves and intends to address the complex configuration in which breaking events appears during the course of the wave evolution. The interesting idea used in the present work is to combine a classical non-breaking propagation scheme (here based on MNLS) with machine learning to account for the effects of wave breaking. The training data are based on experiments in water tanks. This original work is of great interest to the nonlinear water wave community since the accurate and efficient description of breaking sea states is still an open question. The application of the methodology developed can for sure be extended to other fields of physics. Then, I would say that the work fits the scope of Nature Communications.

However, the paper is in my opinion not suitable for publication in its present form. My main concern is related to the lack of generality of the results presented. The training data sets are specific to the cases tested (namely Wave Category I and Wave Category II) and it is consequently not obvious that the present methodology is usable in a practical configuration in which one does not know what kind of breaking will appear in the sea state. A so-called external data is presented to partly address this issue but even if the scale is different as well as the spectrum shape, it is still a focused event. Dynamics of a focused wave will be very similar and even if it brings new insight, it is for me not sufficient to be sure of its possible generalization to any kind of sea states.

We thank the reviewer for their constructive comments. We would like to emphasize that wave breaking is a very complex problem, and one needs to understand the fundamentals of deterministic cases before moving on to random seas. The deterministic waves (Wave Category I modulated plane waves and Wave Category II dispersively focused irregular waves) are selected to take on this first step. Here, breaking can be detected with certainty and precision in the training data itself (the laboratory experiments), allowing for the clearest possible assessment of whether our ML model works.

Following the reviewer's remark, we have conducted new wave tank experiments for a unidirectional random sea state. That is, a JONSWAP spectrum with random phases. We have added Wave Category III: random irregular waves to the manuscript. As the signature of breaking is not evident in the spectrum, we only compare to the time-domain evolution. We show that our algorithm can successfully correct the unphysical amplitudes predicted by the MNLS for irregular waves.

Having demonstrated that the wave breaking process can be decoupled from potential flow as long as memory is included, we envision our method can indeed be applied to any measurement-model combination. To sketch the path we foresee towards a more general application of our work to realistic sea states we have added a paragraph to the Discussion section, and we answer in more detail on the Reviewer's next question on the salient point to keep from this work for practical application.

The major points I can indicate that may help the authors to improve their manuscript (and detail my previous statement):

Looking at the training data sets characteristics and the one tested in Sec. 2.1 and 2.2, the simulated waves are probably very close to one original data set. Then, even if it is an interesting preliminary step to test the procedure, additional test cases are necessary to me.

We understand that the presentation of more test cases is more insightful. We have added an overview of the other test cases in Supplementary Information sections 2 (figures SI 2.3, SI 2.4) and 3 (figures SI 3.3, SI 3.4) for wave categories I and II, respectively. The average MSE of the MNLS prediction and the MNLS-FMLC for both wave categories is added in the main text as a new Figure 5. Finally, we emphasize in the Methods section that the network is never trained on the full propagation length, only on propagation segments. In the presented test cases, we compare to the maximum length of information available, i.e., to the full length of the tank, starting at the first wave gauge.

The lack of generality of the work is partly pointed out by the authors in Discussion section. However, they point toward another methodology based on infinitesimal-domain method to possibly solve the issue. Then, what are the salient point one can keep from this work if it needs to be modified for practical application?

The main new insight this work brings is that that the physics of breaking can be decoupled from a potential-flow model. With this pivoting knowledge, moving forward we envisage two possible paths that can be pursued in parallel. We distinguishing those that make use of the infinite-domain and those that make use of the finite-domain approach. In the finite-domain approach, the same approach as in this paper can readily be used for other combinations of models and measurement techniques, depending on the purpose of the study, such as deterministic wave forecasting, statistics, or extreme event detection. For each purpose the blended approach as in this work can be applied: a model that captures the basic physics without breaking, a machine learning layer with memory, and measurements that track the evolution at finite intervals. The challenge, especially in two-dimensional sea states, will be to obtain sufficient experimental data.

For the infinitesimal-domain method, the first challenge in future work will be to identify and develop suitable, high-fidelity models that capture at least the most salient features of wave breaking, are able to do so for a range of wave types and conditions and can generate a sufficient amount of training data without excessive computational cost. In addition, for this method, it will be crucial to guarantee stability and boundedness.

We have adapted the Discussion to include this point.

Methods - I am not sure to have understood in details the two RNNs that are referred to in the text and in Figure 6. Does the resolution occur in the real space or in the spectral space? Or are both the resolution done at the same time? If so, how is ensured the compatibility between both?

The stability of the evolution being able to go from the initial condition at x_i to the desired position x_f is guaranteed by the damped-MNLS equation. As the envelope is a complex function in this evolution, the compatibility is ensured.

Then, after this finite stretch of evolution, we employ two different RNN's to merely apply a correction. One on the time domain, and one on the spectral domain. To go from spectrum to time series and visa versa as to ensure compatibility, one would need the phase information, which we cannot obtain from the experiment (see Supplementary Information Section A).

An option is to only focus on the time domain evolution, one would lose the spectral information. Nevertheless, it is often useful to have the spectral information to understand the wave properties that do not require phase information (spectral peak, Stokes drift, Benjamin Feir Index etc.). The solution is to learn a separate evolution in Fourier space. Indeed, in optical fibers, the same method of having two networks for both the real and spectral space evolution of the NLS is applied, see Salmela 2021.

While there is no guarantee for compatibility between both domains, the correction is only a small variation on the main dynamics set down by the MNLS. In addition, for training, the complex envelope is available at each wave gauge position, and therefore the spectrum and time series are compatible at these points, so the RNN will in a sense learn to produce compatible solutions.

We have clarified this point in the Methods section.

I.197: it is explained that bound modes are filtered from the measurement signal: how is it done practically for a broad-banded spectrum such as JONSWAP?

*For both wave types, the signal is filtered beyond a cut-off frequency $f_c = 2*f_p$, where f_p is the peak frequency of the JONSWAP spectrum. While the JONSWAP spectrum is theoretically broad-banded, in a way our filtering process is facilitated by the limit of the wave maker, which is at 2 Hz, with an f_p between 1-1.5 Hz, this cuts off a large part of the bound modes at the wave generation. The agreement between the evolution of the MNLS prediction and the data for non-breaking waves shows that the nonlinear interaction the bounds modes could have with the modes $<2*f_p$ do not influence the evolution significantly.*

Appendix B.1: How is it possible that MNLS is working worse than MNLS-FDML in a non-breaking wave condition? I think it needs to be discussed briefly.

Firstly, the MNLS is a very good approximation for a non-breaking wave, but it does not include all the physics involved in the real experiment. For instance, the experiments are not in truly infinitely deep water, have sidewall friction, reflections from the beach, and a broader bandwidth than the simulation. In contrast, the FDML correction is trained to learn the discrepancy between the actual measurement and the MNLS simulation, whether that is breaking or other inaccuracies. Therefore, it can outperform the MNLS also on non-

breaking cases. Secondly, the FDML correction is optimized based on the MSE. However, especially in real space, the MSE is not a perfect metric for the quality of a prediction, as it severely penalizes a phase shift.

We have now added this as a brief discussion in Supplementary Information 2.1.

Appendix C.1: it is said that FDML applies over-damping in this non-breaking condition. I think this is an issue toward the possible generalization of the use of the method. Is it possible to enhance this with additional training data (non-breaking)? Or do you expect that those additional data will alter the results presented in breaking conditions?

We indeed mention that the FDML applies a slight over-damping. The reviewer is correct that the over-correction is likely due to not having seen enough non-breaking cases. Additional data should not alter the results for the breaking conditions because these extra samples will just extend the distribution of 'all possible waves' that is being sampled by the training data, instead of it moving away from breaking cases. Even in extreme cases of strongly imbalanced data sets, special techniques can be adopted that guarantee a uniformly accurate/balanced training, see Chawla 2005.

References used in this reply:

Salmela, L., Tsipinakis, N., Foi, A., Billet, C., Dudley, J. M., & Genty, G. (2021). Predicting ultrafast nonlinear dynamics in fibre optics with a recurrent neural network. *Nature Machine Intelligence*, 3(4), 344–354. <https://doi.org/10.1038/s42256-021-00297-z>

Chawla, N. V. (2005). Data Mining for Imbalanced Datasets: An Overview. In O. Maimon & L. Rokach (Eds.), *Data Mining and Knowledge Discovery Handbook* (pp. 853–867). Springer US. https://doi.org/10.1007/0-387-25465-X_40

Reviewer #3 (Remarks to the Author):

We would like to thank the reviewer for their positive response and helpful detailed comments, which we address individually below. Our response to the reviewer's report reproduced below is in italics and indented.

The authors present the very interesting problem of predicting wave breaking at high resolution, at a low computational cost, using hybrid physics and machine learning (ML) techniques. A noteworthy aspect is that the authors have used experimental data instead of numerical simulation data that is traditionally used in such studies. This adds to the complication since the experiments provide data only at specific spatial locations with gauges, as opposed to the entire field (which numerical simulations can, but are very expensive).

As a solution, the authors propose a hybrid ML + physics approach that they claim exploits both the Modified nonlinear Schrodinger (MNLS) equation while using a neural network to "correct" it with experimental data. While the motivation and overall direction is outlined well, I unfortunately had a very difficult time a) Extracting the actual *details* of the method that is proposed, and b) In seeing the novelty in this work.

Figure 6 is helpful, but not enough for the detailed mathematical content for a Nature publication, as it is. In Pg. 2, Lines 68-70 the authors say "...applies an RNN". However, there is no exact information on what makes "FDML" any different from just feeding data to an RNN. In fact, there are plenty of results shown, but no information about how "MNLS + FDML" model is developed! In fact, the only relevant section is the one on LSTM and Eqn. (7), which makes it very confusing to understand exactly how the MNLS "interfaces" to the LSTM and why it is a novel enough approach to warrant publication in Nature Comms, since there are several papers that have already demonstrated some flavor of this (Too many to cite here). While I can see an innovation being on how they use experimental as opposed to simulation data, this aspect alone is not enough to carry the whole paper, since the rest of the method is not explained clearly. For this reason, *I am forced to not recommend publication of this manuscript in its Current form*.

However, I highly encourage the authors to resubmit so the algorithm and innovation is clearly outlined, with the appropriate citations to relevant, related works already in this field.

We thank the reviewer for their constructive comments.

Regarding the first comment on the details of the method: we thank the reviewer pointing out that this part should indeed also be included in the manuscript itself, and that provided code is no replacement for this. We have added a detailed algorithm as Supplementary Information 6, and have clarified the method in the Method section of the main text. We will address this point further in the specific questions below.

Regarding the second comment on the novelty of the work: the main novelty is in the physics that are revealed: a priori it is not obvious that a nonlinear ML function with memory could at all capture the turbulent breaking process in a tractable way. This result allows to decouple a potential-flow model and wave breaking. For deterministic wave forecasting, wave breaking has been a wall for progress. In this context, our result is a huge insight, and a breakthrough that now paves the way of using the recipe of A) simple wave propagation model and B) ML layer on a multitude of new model and measurement combinations.

*Regarding the method itself, as the reviewer points out, as the ground truth is provided by experimental observations, and thus the method required a different frame of mind than when a solution is known, as we shall also discuss in the point about physical constraints below. Training any network architecture indeed in the end comes down to feeding (input, output) pairs to it (again some sophistication can be implemented through constraints or optimizer). It is the **combination** of the physics with the ML method that is crucial, as our first ventures into using the common methods for synthetic data illustrates, because of the phase information problem outline in SI section 1. For instance, we also tried several other architectures, e.g. a hybrid method where the phase was retained from the MNLS, but the absolute value was corrected to the measurements by the RNN; this yielded unphysical results. Therefore, the choices of what can actually be used as input and output, to still end up with a usable initial value solver are non-trivial and depend heavily on the specifics of the physical problem.*

Once that was clearly outlined, the creation of training pairs is not straight-forward. Simply using the entire experiment would not yield enough data. We therefore augmented the data by creating smaller propagation segments different evolution trajectories from the same experiment, depending on the starting point (see figure SI 6.1 for an illustration). Finally, obtaining the necessary amount of experimental data is a novelty. In our estimation, the typical number of experiments in the best field journals in fluid mechanics is of the order of 10, as it is so time consuming to obtain these. Here we performed over 400 experiments, overheating the wave maker several times.

Some specific questions:

1) Where do $|a_MNLS|$ and $|a_true$ come from? Lines 239 and 240 give some indication, but this is a very important part of the authors' contribution, and must be explicitly defined in an equation and/or figure for clarity. From the information provided in the manuscript, I fail to understand the novelty in ML, when the approach just uses a standard LSTM. I would *guess* that the innovation is somewhere in how the MNLS and LSTM are interlinked, but I cannot find that in the paper.

Indeed this belongs to the novelty of the present paper. We have adapted the description in the Methods and in SI 6 we have now detailed the algorithms necessary to obtain the training pair $\{|a_MNLS|, |a_true|\}$.

2) The authors make no comment on the stability or boundedness of the trajectory predicted by the LSTM, apart from the spline interpolation performed on the gauge data. While 12 gauges are enough for this case according to the authors, it provides no scientific basis that this method is generalizable beyond this specific experimental dataset. Additionally, LSTMs have no implicit boundedness guarantees, so there is little reason for the situation in Fig 6(c) to hold true.

The reviewer is right that the stability and boundedness is critical when using a blended method for a dynamical system. In an infinitesimal domain (or differential) correction, where one step builds on the deviation of the true trajectory of the previous step, the correction as depicted in -now- Fig 7 (b) should indeed be insured.

However, in our case, the finite-domain correction, the MNLS is solved over the entire propagation stretch. As the MNLS integration through the split-step Fourier method for non-breaking waves is a tried-and-tested stable method. Introducing the 3rd order damping term in equation (1), quadratically depletes higher modes in the spectrum, and ensures numerical stability even for very steep waves, while not affecting the evolution. The subsequent RNN correction is a relatively small, although physically important. Of course, this correction can still be inaccurate such that Fig 7 (c) does not hold true. Note that a stable infinitesimal blended solver that adheres to physical constraints such as conservation of energy can also still predict an inaccurate trajectory / evolution. As we will address in the point below, there are no conservation laws nor evolution equations for breaking surface waves to bound the solution.

We also note that we use the word trajectory, but the MNLS is a PDE, so one must keep in mind that instead of describing the evolution of isolated trajectories as in coupled ODEs, the envelope a is a continuous variable in the time (as we propagate in space).

Regarding the generalizability of the method beyond this dataset and the spacing of the wave gauges.

The Nyquist–Shannon sampling theorem provides a lower bound for the sampling frequency in both space and time, allowing to capture all the information from a continuous signal of final bandwidth. The sampling frequency must be twice that of the highest frequency in the continuous signal.

In the time domain, we need to resolve the surface elevation first, as we need that to obtain the complex envelope. The highest frequency in our experimental data is $f_0 = 1.7\text{Hz}$, $k_0 = 11.6$, $\lambda = 1.85\text{m}$. The highest frequency we want to observe up to the first harmonic, that is $2*f_0$. Therefore, the sampling frequency should be double that: $f_s = 2 * 2 * f_0 = 6.8\text{ Hz}$. The sampling frequency of our wave gauges is 100nHz so this is more than enough.

For the evolution in the propagation direction, we are interested in the envelope. We assume that in order for the slowly varying envelope approximation to hold, one period of the envelope covers N periods of the surface elevation. It is difficult to assign an exact number to this for different wave types, and for irregular waves this depends on the spectral width, but let us say $N=5$. i.e $\lambda_{env} = 5*\lambda_0$. Then, if we want to resolve features of this frequency, wave gauge spacing should be: $5*1.85/2 = 4.625$. Since the wave gauges are placed at fixed physical distance (about 3 meters) for wave categories I and II, some lower frequency runs are over-sampled. Another way to verify if the signal is sufficiently sampled is by comparing the non-breaking (MNLS) simulation with the experimental data, as done for instance in SI Section 1.

3) How do you ensure that physical constraints are obeyed? The LSTM loss functions are “soft” constraints as they merely act as a suggestion to the optimizer, but do not guarantee any fidelity to constraints - There are papers on “hard” constraints that do this job and are much more robust and accurate than soft constrained neural networks (I request the authors to read those papers - there are many and I don’t want to bias by providing my preferred references) and these approaches are the current state of the art!

In our discussion we tried to bring to the attention that using synthetic data - where the full solution, or PDE that describes the physics (Burgers, NLS, etc) is known- in the ML community brings along a certain mindset that does not always transfer to real-world problems. Indeed, as the reviewer suggests, in this case, one can use these known PDEs that were used to generate the ground truth in the first place as constraints of the evolution of a reduced dynamical model. Alternatively, if one has information on the physical or mathematical constraints such as conservation of energy, mass, momentum, a probability density integrating to unity, these can be implemented as hard or soft constraints, making the system more robust and training faster or with less data.

However, in our case, and in many cases where experiments serve as the ground truth, we are trying to capture unknown physics in an evolution model. There is no known, more complicated, PDE that the solution must obey, as wave breaking cannot be captured by (nonlinear) term in the equation, because physically wave breaking cannot be described by a potential flow equation. Regarding physical constraints, unfortunately, the surface elevation of breaking waves does not conserve energy, nor symmetry in the spectrum (the ratio of the norm and momentum). We cannot even say that the energy decay must be smooth, as it can be of timescales much shorter than the slowly varying envelope.

The only constraint we could think of was that the output should be positive, but this is fulfilled by the activation function of the output neurons (ReLU). We are therefore in a way saved by the robust MNLS that can solve over the entire propagation domain, even if breaking occurs along the way due to the viscosity damping terms. We have now clarified this in the manuscript when explaining the cost function (equation 9).

Nevertheless, as mentioned before, in an infinitesimal correction setup, which we intend to pursue in further work such constraints to ensure stability will be very relevant. In this case, the constraint of not letting the norm increase will likely be important. If a physics-based model that concentrates on the surface elevation directly would be used, we can also think of constraints on the velocity field. Thanks to the reviewer we have familiarized ourselves with the state of the art in the literature. In the discussion section, we have elaborated on how this is a critical point to think of for future work on the infinitesimal method and have added the following references on hard constraints:

Mohan, A. T., Lubbers, N., Livescu, D., & Chertkov, M. (2020). Embedding Hard Physical Constraints in Neural Network Coarse-Graining of 3D Turbulence. *ArXiv - Preprint*, 1–13. <http://arxiv.org/abs/2002.00021>

Dener, A., Miller, M. A., Churchill, R. M., Munson, T., & Chang, C.-S. (2020). Training neural networks under physical constraints using a stochastic augmented Lagrangian approach. *ArXiv - Preprint*, 1–21. <http://arxiv.org/abs/2009.07330>

Balogh, B., Saint-Martin, D., & Ribes, A. (2021). A Toy Model to Investigate Stability of AI-Based Dynamical Systems. *Geophysical Research Letters*, 48(8). <https://doi.org/10.1029/2020GL092133>

Beucler, T., Pritchard, M., Rasp, S., Ott, J., Baldi, P., & Gentine, P. (2021). Enforcing Analytic Constraints in Neural Networks Emulating Physical Systems. *Physical Review Letters*, 126(9). <https://doi.org/10.1103/PhysRevLett.126.098302>

Li, Z., & Ravela, S. (2019). *Neural Networks as Geometric Chaotic Maps*. 1–7. <https://doi.org/10.1109/tnnls.2021.3087497>

Finally, In my opinion, the goal of a research manuscript is to provide enough detail for someone to implement their technique from scratch. While making the code available is a commendable move, it is not a substitute for the actual text. There is indeed a possibility that there indeed is something very unique in the MNLS + FDML algorithm that I might have missed due to lack of relevant details from the authors, but I cannot see it from the content in the paper. However, readability is very important to readers who may not be from a specific field (wave breaking or machine learning), and is severely lacking for those outside this specific niche. That said, I would be open to reassessing the paper on its technical merits once the requisite details are added.

We thank the reviewer for pointing this out. We now realize that this lacking information may have compromised the readability of the manuscript. We have clarified the algorithm in the Methods section, and refer to the Supplementary Information section 6 for a full description of the algorithm. We expect that we have added sufficient detail and that the text is readable for a broader public.

REVIEWERS' COMMENTS

Reviewer #1 (Remarks to the Author):

General Impression: I have read the authors' response to all three reviewers and the revised manuscript with its supplementary material. There were some shared questions across the reviewers, and in general, I believe the authors have addressed my comments. They have clearly performed and included substantial additional analysis and explanations. Below, I briefly restate my initial review points and assess the authors' revisions and new material. All my minor comments were addressed.

Validity: Include some overall performance statistics to demonstrate that the example cases are representative.

- The new Figure 5 and Supplementary section 2 and 3 satisfy my request. I appreciate the authors' explanation of the proportions of the data used for training, validation, and testing. Regardless of the smaller percentage of cases used for testing, I think the summary statistics in Figure 5 help illustrate overall performance in a clear way.

Significance: How far is the current method from applicability to more realistic wave fields?

- The authors added a "Wave Category III: random irregular waves". These are unidirectional JONSWAP spectrum waves with random phases. This is a step in the right direction and provides an opening in the manuscript text for the authors to discuss future research directions in more detail. While another reviewer seems to take more issue with the fact that this work isn't directly applicable to real-world waves, I think it is standard practice to prove a new method on simplified cases first.

Data & Methodology: Add more context for why certain terms are included in the MNLS equation and others are excluded for this work. Comment on how those choices might change for real-world application of the method.

- The authors have added sufficient explanation in the methods section to address my comment. They also added the equations that another reviewer requested. Taken together, I think these revisions will help a reader understand the motivation and choices made in the physical model, and to see those choices explicitly in equation form.

Data & Methodology: Describe the quality or resolution of data that would be required for the infinitesimal domain method for broad-banded waves.

- The authors mostly address this in their response to reviewer rather than in the text, though they do add some to the manuscript's Discussion section. That's fine with me. It is a bit outside the scope of the paper. I appreciate what was provided in their response to my question.

New minor comment: I found several small typos throughout the revised manuscript, particularly in the new text sections. The final manuscript should be carefully proofed to remove these errors.

"Reviewer 1"

Roxanne J Carini, PhD
Senior Oceanographer, NANOOS
Applied Physics Laboratory,
University of Washington

Reviewer #2 (Remarks to the Author):

At first, I would like to thank the authors for the work done since last review and for the answers provided to my initial comments. I think that the paper has been nicely enhanced and that it is now clearer and more complete.

This work is, in my opinion, an important first step toward the use of machine learning in the context of nonlinear water wave propagation. This presents a plausible alternative to the recent development of different breaking models for nonlinear potential flow solvers. Then, I am open to publication of this work once the following comments are taken into account.

Section 2.3: The random irregular waves is a nice addition to the paper. However, I have a question regarding the training data for this Wave Category III. From what I understand those data corresponds to a JONSWAP spectrum with $\gamma=3.3$ and different significant wave heights? I think it should be made clearer if all those data are used at the same time for training data or if one particular significant wave height is used as training data for the FDML used to model this specific significant wave height? If this is the case, this is a serious limitation that needs to be commented. In this new section, the scale effects are not addressed or comments: do you expect any difficulty in applying the same model to another scale? Those scale effects are partly addressed in SI 3.4 but it may be worth mentioning if for Wave Category III similar behaviour is expected.

Figure SI 1.1: presenting unwrapped phases may be misleading in the differences observed, even if I agree that there is no perfect way of presenting phase differences

SI 3.1: it is said here that 'FDML correction does not apply a correction in the non-breaking case'. This is contradictory with a statement in previous section on the fact that 'other inaccuracies' are taken into account in the FDML. Could you clarify? In addition, in Fig. SI 3.1, can you explain where does the difference between experiment and MNLS come from. It seems that there is quite some dissipation in this non-breaking case.

SI 4. One obvious advantage of FDML is that no parametrization of the model is necessary once it is established. However, the choice of beta value for Kato & Oikawa model has probably a significant influence on MSE. Then, does the choice $\beta=0.2$ for SI 4.1 minimize the MSE at a particular location? For fair comparison, I think such approach should be carried on.

Minor:

l.198 through instead of though?

l.245: a piston wave maker

Eq.(2): the envelope should be dependent on x ?

Reviewer #3 (Remarks to the Author):

The authors have done a great job of addressing all my concerns and making the paper more detail-rich. The addition of copious supplemental information is also welcome. I fully recommend this paper for publication in its current form.

Reviewer #1 (Remarks to the Author):

[previous remarks deleted for brevity]

- New minor comment: I found several small typos throughout the revised manuscript, particularly in the new text sections. The final manuscript should be carefully proofed to remove these errors.

We thank the reviewer for pointing out these typos. We fixed these in the resubmitted manuscript.

Reviewer #2 (Remarks to the Author):

At first, I would like to thank the authors for the work done since last review and for the answers provided to my initial comments. I think that the paper has been nicely enhanced and that it is now clearer and more complete.

This work is, in my opinion, an important first step toward the use of machine learning in the context of nonlinear water wave propagation. This presents a plausible alternative to the recent development of different breaking models for nonlinear potential flow solvers. Then, I am open to publication of this work once the following comments are taken into account.

- Section 2.3: The random irregular waves is a nice addition to the paper. However, I have a question regarding the training data for this Wave Category III. From what I understand those data corresponds to a JONSWAP spectrum with $\gamma=3.3$ and different significant wave heights? I think it should be made clearer if all those data are used at the same time for training data or if one particular significant wave height is used as training data for the FDML used to model this specific significant wave height? If this is the case, this is a serious limitation that needs to be commented.

All the different significant wave height cases are used for training, subsequently, the FDML can be applied to any wave height. We only had the chance to test the wave heights we had available from the experiment, but we presume that due to the scaling we apply there will be no issues in using unseen wave heights. See reviewer's next sub-question

- In this new section, the scale effects are not addressed or comments: do you expect any difficulty in applying the same model to another scale? Those scale effects are partly addressed in SI 3.4 but it may be worth mentioning if for Wave Category III similar behaviour is expected.

In the discussion, we now comment that since we scale the wave data by its frequency and initial amplitude, and the model works across the available wave heights, we expect the model to extrapolate well to other carrier wave frequencies and wave heights.

- Figure SI 1.1: presenting unwrapped phases may be misleading in the differences observed, even if I agree that there is no perfect way of presenting phase differences

We have now noted this comment in SI 1.

- SI 3.1: it is said here that 'FDML correction does not apply a correction in the non-breaking case'. This is contradictory with a statement in previous section on the fact that 'other inaccuracies' are taken into account in the FDML. Could you clarify? In addition, in Fig. SI 3.1, can you explain where does the difference between experiment and MNLS come from. It seems that there is quite some dissipation in this non-breaking case.

For modulated plane waves the 'other inaccuracies' in the non breaking case were systematic enough to give the FDML a better prediction. For the focused irregular waves, the MNLS does not seem to have the same systematic deviation from the measurement, except over-estimating the maximum value. The FDML therefore has less 'other inaccuracies' to correct. We apologize for the confusion, we now add that there are no significant corrections, especially in the spectrum. We already mention that there is a slight over-damping as compared to the experiment.

The fact that the MNLS over-estimates the maximum value so much in comparison to the experiment could be due to the fact that the dispersion in the experiment differs from that in the simulation, and therefore the modes do not constructively interfere perfectly to add up to the theoretical maximum. Another reason could be that the waves are damped more by the sidewalls, which is not taken into account in the simulation.

- SI 4. One obvious advantage of FDML is that no parametrization of the model is necessary once it is established. However, the choice of beta value for Kato & Oikawa model has probably a significant influence on MSE. Then, does the choice beta=0.2 for SI 4.1 minimize the MSE at a particular location? For fair comparison, I think such approach should be carried on.

We have tried the range of beta values and have chosen the value that gave the best overall result to give a fair comparison. As we explain in the text, if beta was set too low there is no effect on the evolution. When set too high, the wave simply disappears at a short evolution distance. We tried to find a middle value that decreased the MSE at the breaking location and thereafter, while not having too much dissipation to the extent that the wave disappeared.

- Minor:
l.198 through instead of though?
l.245: a piston wave maker
Eq.(2): the envelope should be dependent on x?

We thank the reviewer for spotting these errors. We have corrected them.